# A forebrain hub for cautious actions via the midbrain

**Ji Zhou†, Muhammad Sarmad Sajid†, Sebastian Hormigo, Manuel A Castro-Alamancos***

Department of Neuroscience, University of Connecticut School of Medicine, Farmington, United States

## eLife Assessment

This **valuable** study uses fiber photometry, implantable lenses, and optogenetics, to show that a subset of subthalamic nucleus neurons are active during movement, and that active but not passive avoidance depends in part on STN projections to substantia nigra. The strength of the evidence for these claims is **solid** and this paper may be of interest to basic and applied behavioural neuroscientists working on movement or avoidance.

*For correspondence:
mcastro@uchc.edu

†These authors contributed equally to this work

Competing interest: The authors declare that no competing interests exist.

## Abstract

Adaptive goal-directed behavior requires dynamic coordination of movement, motivation, and environmental cues. Among these, cautious actions, where animals adjust their behavior in anticipation of predictable threats, are essential for survival. Yet, their underlying neural mechanisms remain less well understood than those of appetitive behaviors, where caution plays little role. Using calcium imaging in freely moving mice, we show that glutamatergic neurons in the subthalamic nucleus (STN) are robustly engaged by contraversive movement during cue-evoked avoidance and exploratory behavior. Model-based analyses controlling for movement and other covariates revealed that STN neurons additionally encode salient sensory cues, punished errors, and especially cautious responding, where their activity anticipates avoidance actions. Targeted lesions and optogenetic manipulations reveal that STN projections to the midbrain are necessary for executing cued avoidance. These findings identify a critical role for the STN in orchestrating adaptive goal-directed behavior by integrating sensory, motor, and punitive signals to guide timely, cautious actions via its midbrain projections.

## Introduction

Adaptive behavior often requires animals to take goal-directed actions in response to environmental cues that predict rewards or potential threats (*Thorndike, 1898*). The ability to initiate a response rapidly enough to prevent harm—yet not so prematurely that it leads to unnecessary risk—defines cautious behavior (e.g. vacillating before crossing the street). This form of behavioral control has been extensively studied through speed-accuracy trade-off tasks and evidence accumulation models, yet much of this work has focused on appetitive outcomes—such as the presentation or omission of reward—rather than the prospect of harmful, aversive consequences (*Smith and Ratcliff, 2004*; *Gold and Shadlen, 2007*; *Bogacz et al., 2010*; *van Maanen et al., 2011*; *Guitart-Masip et al., 2012*; *Heitz and Schall, 2012*; *Ratcliff and Frank, 2012*; *Yee et al., 2022*; *Zhou et al., 2022*). Understanding how the brain coordinates cautious decision-making under threat is critical for uncovering the neural mechanisms that guide adaptive, motivated actions.

The STN, located within the subthalamus alongside the GABAergic zona incerta, is primarily composed of glutamatergic neurons. It is a key component of the basal ganglia's indirect pathway,

interconnecting with the globus pallidus externa (GPe) and projecting to the basal ganglia output nuclei, including the substantia nigra pars reticulata (SNr) in the midbrain. The STN integrates a diverse array of inputs from both forebrain and midbrain regions and provides a hyperdirect pathway from the cortex to the midbrain, bypassing the striatum (*Kita and Kitai, 1987*; *Albin et al., 1989*; *Canteras et al., 1990*; *DeLong, 1990*; *Gerfen and Wilson, 1996*; *Smith et al., 1998*; *Nambu et al., 2002*; *Kita and Kita, 2011*; *Wilson and Bevan, 2011*; *Prasad and Wallén-Mackenzie, 2024*). The role of the STN in self-paced movement and action control is well-documented (*Isoda and Hikosaka, 2008*; *Bonnevie and Zaghloul, 2019*; *Klaus et al., 2019*), and it is a common deep brain stimulation (DBS) target for treating Parkinson's disease (*Limousin et al., 1995*; *Gittis and Sillitoe, 2024*). It is also targeted for treating obsessive-compulsive disorder (OCD), which is characterized by abnormal, repetitive actions often triggered by external cues that the subject cannot inhibit (*Mallet et al., 2008*; *Haber et al., 2021*). However, there are divergent perspectives on STN's function during movement and action generation. Intriguingly, studies in humans suggest that STN is involved in action slowing or cautiousness in the face of conflict or difficulty (*Frank et al., 2007*; *Cavanagh et al., 2014*; *Herz et al., 2024*). It has also been implicated in action cancellation (*Aron, 2011*; *Schmidt et al., 2013*), although other studies propose that different pathways may mediate the stopping of actions (*Mallet et al., 2016*; *Aristieta et al., 2021*; *Bevan, 2021*; *Friedman and Yin, 2023*). Similarly, the involvement of STN in movement control is ambiguous; some studies report that STN activation suppresses movement (*Fife et al., 2017*; *Guillaumin et al., 2021*; *Xie et al., 2022*), while others indicate that it facilitates movement (*Watson et al., 2021*; *Fan et al., 2023*; *Friedman and Yin, 2023*). Moreover, recent recordings from STN neurons in head-fixed mice show activation correlated with self-paced locomotion (*Callahan et al., 2024*), but its activation during cued goal-directed actions where freely moving mice must generate slow onset, cautious responses in adaptive contexts has been less explored.

We recorded and manipulated STN neuron activity in freely behaving mice using cell-type-specific fiber photometry, miniscope calcium imaging, optogenetics, and genetically targeted lesions to investigate their role in exploratory and goal-directed behaviors. Our findings reveal that STN neurons encode contraversive movements and cautious, cued goal-directed avoidance actions characterized by slow onsets and are essential for generating them.

## Results

### STN activity encodes movement in the contraversive direction

To assess the population activity of glutamatergic STN neurons during movement, we expressed GCaMP7f (*Chen et al., 2013*) in these neurons by locally injecting a Cre-AAV in Vglut2-Cre mice (n=8). After implanting a single optical fiber within the STN, we employed calcium imaging fiber photometry, as previously described (*Hormigo et al., 2021a*; *Hormigo et al., 2023*; *Zhou et al., 2023*). *Figure 1A* illustrates a representative trajectory of the optical fiber targeting GCaMP-expressing glutamatergic STN neurons. The estimated imaged volume extends ~200 μm from the optical fiber ending, encompassing ~$2.5 \times 10^7$ μm$^3$ (*Pisanello et al., 2019*).

In freely moving mice, we conducted continuous measurements of calcium fluorescence (ΔF/F) and spontaneous movement while mice explored an arena. Cross-correlations were computed between movement and STN activation to relate these continuous variables (*Figure 1B* upper). Overall movement was strongly correlated with STN neuron activation. When the movement was dissociated into rotational and translational components, the cross-correlation predominantly involved the translational movement. A linear fit between the movement and ΔF/F (integrating over a 200 ms window) revealed a strong linear positive correlation between the STN activation and translational or rotational movement (*Figure 1B* lower). These relationships were absent when one variable in the pair was shuffled (*Figure 1B*, lower).

To further evaluate the relationship between movement and STN activation, we detected spontaneous movements and time-extracted the continuous variables around the detected movements peaks (*Figure 1C*) following the same procedures we used in other brain regions (*Hormigo et al., 2023*; *Zhou et al., 2023*). The detected movements were classified into three categories. The first category includes all peaks (*Figure 1C* black traces), which revealed a strong STN neuron activation in relation to movement. The second category includes movements that had no detected peaks 3 s prior (*Figure 1C* red traces), which essentially extracts movement onsets from immobility. This revealed

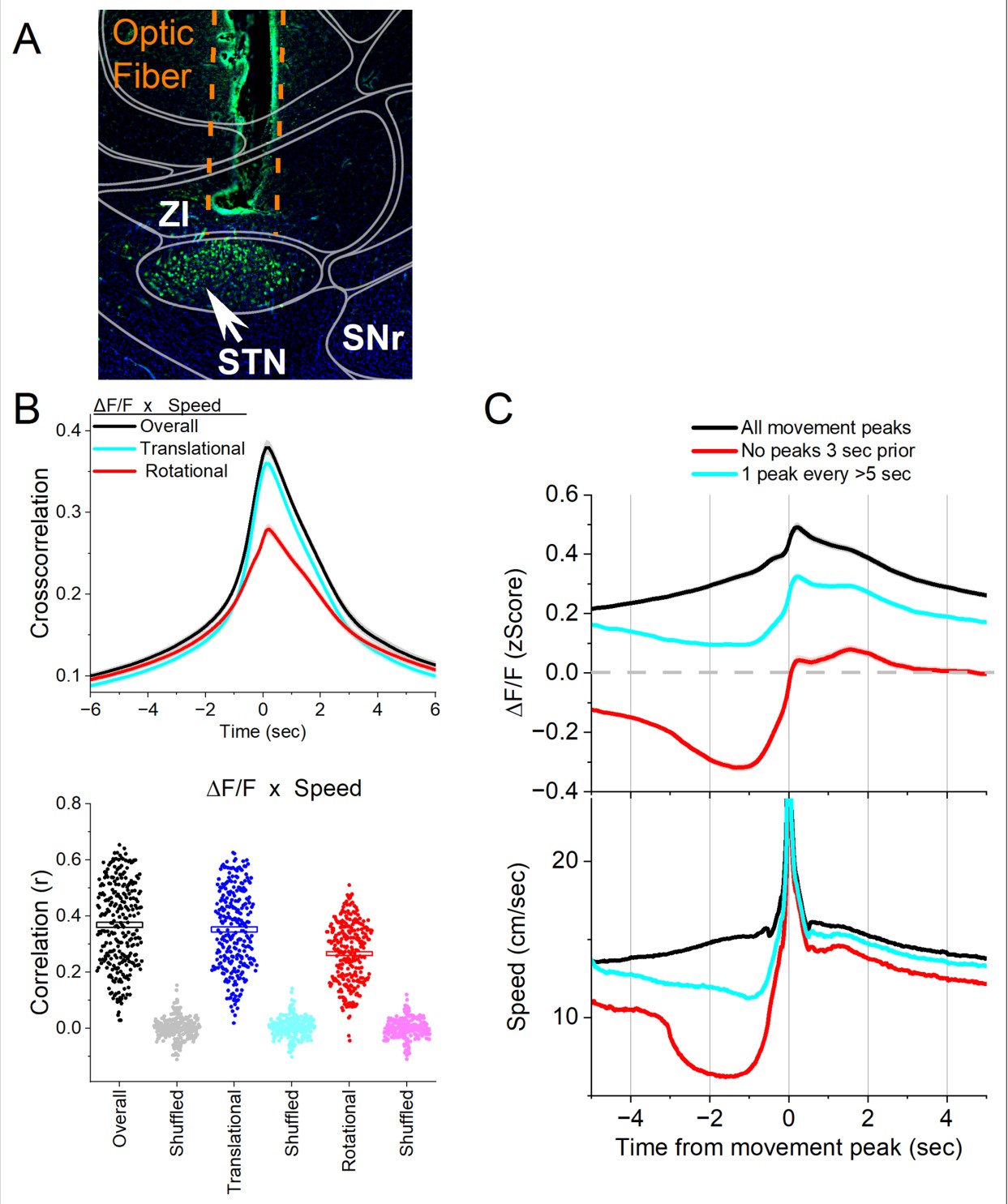

**Figure 1.** Calcium imaging fiber photometry reveals that subthalamic nucleus (STN) glutamatergic neurons activate during spontaneous exploratory movement. (**A**) Parasagittal section showing the optical fiber tract reaching STN and GCaMP7f fluorescence expressed in glutamatergic neurons around the fiber ending. The section was aligned with the Allen brain atlas. ZI, zona incerta; SNr, substantia nigra pars reticulata; STN, subthalamic nucleus. (**B**) Cross-correlation between movement and STN ΔF/F for the overall (black traces), rotational (red), and translational (cyan) components (upper panel). Per session (dots) and mean ± SEM (rectangle) linear fit correlation (**r**) between overall movement and STN ΔF/F, including the rotational and translational components (lower panel). The lighter dots show the linear fits after scrambling one of the variables (lower panel, shuffled). (**C**) ΔF/F calcium imaging time extracted around detected spontaneous movements. Time zero represents the peak of the movement. The upper traces show ΔF/F mean ± SEM of all movement peaks (black), those that had no detected peaks 3 s prior (red), and peaks taken at a fixed interval >5 s (cyan). The lower traces show the corresponding movement speed for the selected peaks. All traces in the paper are mean ± SEM. n=8 mice.

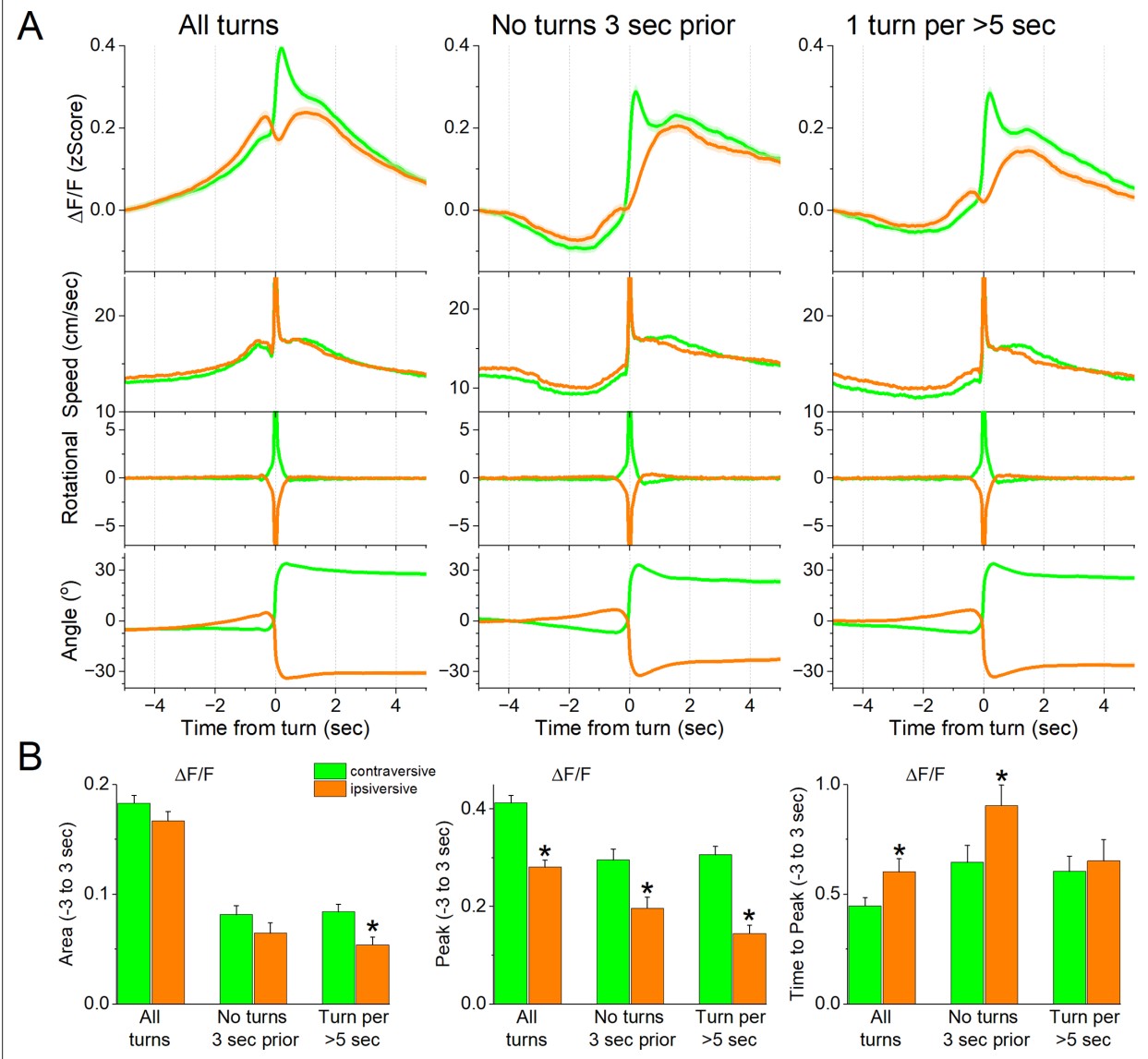

**Figure 2.** Subthalamic nucleus (STN) glutamatergic neurons code the direction of spontaneous contraversive exploratory turning movements. (**A**) ΔF/F calcium imaging, overall movement, rotational movement, and angle of turning direction for detected movements classified by the turning direction (ipsiversive and contraversive; orange and green) versus the side of the recording (implanted optical fiber). At time zero, the animals spontaneously turn their heads in the indicated direction. The columns show all turns (left), those that included no turn peaks 3 s prior (middle), and peaks selected at a fixed interval >5 s (right). Note that the speed of the movements was similar in both directions (the y-axis speed is truncated to show the rising phase of the movement). (**B**) Population measures (area of traces 3 s around the detected peaks) of ΔF/F and movement (overall, rotational, and translational) for the different classified peaks. Asterisks denote significant differences ($p<0.05$) between ipsiversive and contraversive movements. n=8 mice.

a sharp activation in association with movement onset. The third category sampled the peaks by averaging every >5 s to eliminate from the average the effect of closely occurring peaks (*Figure 1C* cyan traces). This category includes movement increases from ongoing baseline movement (instead of movement onsets from immobility) and showed a strong activation of STN neurons. For the three categories of movement peaks, the STN activation around movement was significant compared to baseline activity (Tukey $p<0.0001$). Thus, STN neurons discharge in relation to the occurrence of movement.

We next determined if the STN activation during movement depends on the direction of the head movement in the ipsiversive or contraversive direction. *Figure 2A* shows movement turns in the contraversive (*Figure 2A* green) and ipsiversive (*Figure 2A* orange) directions versus the recorded STN neurons. While the detected turns were opposite in direction and similar in amplitude, the STN

neuron activation was sharper when the head turned in the contraversive direction. We compared the area, peak amplitude, and peak timing of the ΔF/F activation between ipsiversive and contraversive turns. For *all turns* and *no turns 3 s prior*, the ΔF/F amplitude of contraversive turns was larger (Tukey t(240) = 17.6 p<0.0001 and Tukey t(240) = 5.4 *p*=0.0001) and peaked earlier (Tukey t(240) = 3.22 *p*=0.02 and Tukey t(240) = 3.06 *p*=0.03), while the area did not differ (*Figure 2B*). The results were similar for *1 turn per >5 s* but this category showed a stronger contrast between the peaks in both directions, resulting in a significant difference in both the area (Tukey t(240) = 4.27 *p*=0.002) and peak amplitude (Tukey t(240) = 10.4 p<0.0001) but not the peak timing, which becomes more variable for small peaks. Therefore, STN glutamatergic neurons code movement direction discharging sharply to contraversive turns.

The preceding experiments involved calcium imaging fiber photometry, which integrates the activity of populations of STN glutamatergic neurons. Next, we employed miniscope calcium imaging recordings of individual STN glutamatergic neurons in freely behaving mice (n=5). In these mice, we expressed GCaMP7f by locally injecting an AAV in the STN of Vglut2-Cre mice and implanted a GRIN lens into the STN (*Figure 3A*). We recorded the activity of STN neurons (1030 cells) as mice moved in a cage. The neurons were then classified according to their activation during spontaneous movements and orienting turns.

First, we recorded the ΔF/F activity of individual neurons and identified movement onsets by detecting movements with no turns 3 s prior. The ΔF/F time-series activity of each neuron around these movement onsets was extracted and classified using k-means clustering, revealing three distinct neuron classes. *Figure 3B* illustrates the activation patterns of these classes. Class A neurons, comprising 56.5% of the neurons, showed minimal activation during movement onset. Class B neurons, representing 15.9% of the neurons, were inhibited as movement slowed before onset but did not exhibit sharp activation at onset, suggesting a slight modulation by movement speed. Class C neurons, accounting for 27.6% of the neurons, showed a sharp activation in relation to movement onset.

Second, we identified movement turns and extracted the ΔF/F time-series activity of each neuron around ipsiversive and contraversive turns. The activity difference for each neuron was calculated per session to obtain a directional activation bias, which was classified using k-means clustering into three neuron classes. *Figure 3C* depicts the activation of these classes during ipsiversive and contraversive turns (middle and right panels), along with the directional activation bias (left panel) used for classification. *Figure 3D* (top) shows the first two principal components from the k-means for all the cells. An equivalence test showed no significant difference in turn angles and movement amplitudes across the three classes (F(2,1030)=0.66, *p*=0.55). However, the peak ΔF/F bias amplitudes during turns differed significantly between the classes (*Figure 3D* bottom; F(2,1030)=1086.8, *p*<0.0001). Class C neurons (n=192, 18.6%) exhibited a strong discharge to contraversive movements, significantly stronger than the other classes (Tukey q>30, *p*<0.0001 for Class C vs. Class B or Class A). Class B neurons (n=256, 24.8%) showed greater activation in the ipsiversive direction, but with a much weaker bias compared to Class C. Class A neurons (n=585, 56.6%) displayed minimal activation during turns. Thus, approximately 20% of STN neurons were strongly active during contraversive movements, while about 25% exhibited a weak bias towards ipsiversive movements.

## Tone-evoked STN activity reflects both sensory and motor influences

Our next goal was to examine STN neuron activation across a series of cued avoidance procedures signaled by auditory tones (*Zhou et al., 2022*; *Zhou et al., 2023*) in which mice either initiate movement (actively avoid) or withhold movement (passively avoid) to prevent an aversive US. However, salient sensory events, even when not predictive of contingencies, can elicit movements, such as orienting responses at stimulus onset (*Zhou et al., 2023*). Because STN neurons are robustly activated during movement, as shown above, it is important to disentangle sensory-evoked and movement-related contributions. To address this, we examined tone-evoked STN activity during fiber photometry recordings in freely behaving mice, asking whether tone responses could be explained solely by movement or whether they also contained a distinct sensory component.

During these sessions (52 sessions in 6 mice; *Figure 4*), mice were placed in a small cage (half the size of the shuttle box) and presented with 10 auditory tones of varying salience, defined by sound pressure level (low: ~70 dB; high: ~85 dB) and frequency (4, 6, 8, 12, 16 kHz), delivered in pseudorandom

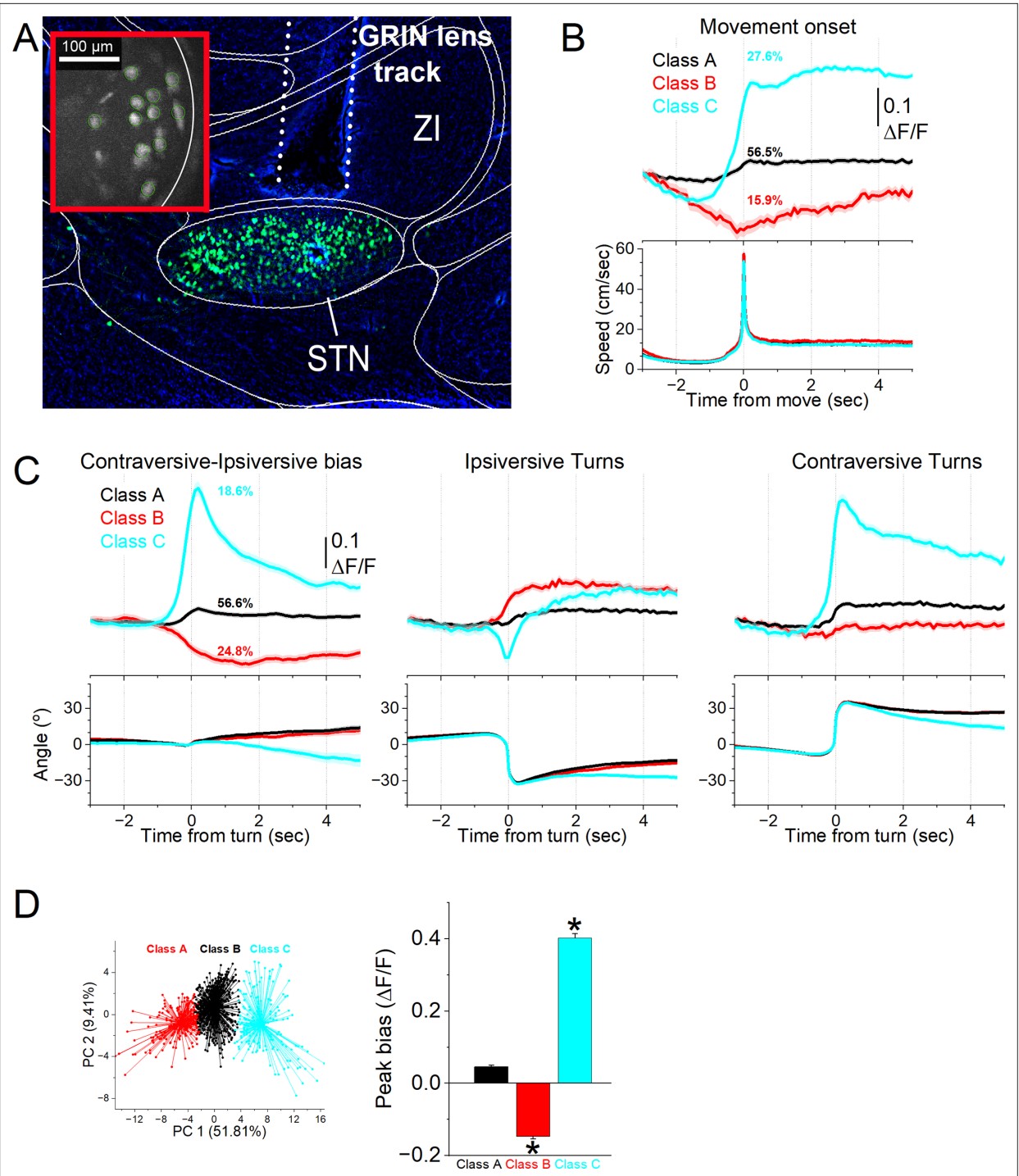

**Figure 3.** A subgroup of STN glutamatergic neurons codes contraversive movements. (**A**) Parasagittal section showing a miniscope GRIN lens tract reaching STN and GCaMP7f fluorescence expressed in STN glutamatergic neurons. The section was aligned with the Allen brain atlas. The red inset shows a FOV of imaged cells during a recording session. ZI, zona incerta; STN, subthalamic nucleus. (**B**) Classification of STN glutamatergic neurons during spontaneous movement onsets with k-means reveals three classes (mean ± SEM). The top traces show ΔF/F calcium imaging, and the bottom traces show the movement onset. Class A neurons activated weakly during movement onset. Class B neurons were inhibited while Class C neurons activated sharply during movement onset. (**C**) Classification of STN glutamatergic neurons during spontaneous turning movements with k-means reveals three classes (mean ± SEM). The top traces show ΔF/F calcium imaging, and the bottom traces show angle of turning direction for detected movements separated by class. The left panels show the activation difference (bias) between contraversive-ipsiversive directions used to classify the cells. The middle and right panels show the corresponding contraversive and ipsiversive movements. Class A neurons did not activate during turns and did not code turn direction. Class B neurons showed stronger activation in the ipsiversive direction. Class C neurons activated more strongly than Class B in the

*Figure 3 continued*

contraversive direction. (**D**) Population comparison of ΔF/F Peak amplitude bias (difference between contraversive-ipsiversive direction) for the three classes of neurons. Asterisks denote significant differences (*p*<0.05) between both directions. The k-means clusters of the three cell classes from C are shown on the top panel. n=5 mice.

order (1 s tones every 4–5 s, each repeated 10 times per session). Tones reliably evoked STN activation as well as movement, with both ΔF/F signals and head speed (overall speed and its rotational and translational components) increasing at higher intensities and frequencies (TwoWayRMAnova, all *p*<0.001). These effects were modest in amplitude (STN: 0.1–0.2 z-score; movement: 2–4 cm/s) but raised the question of whether neural responses reflected auditory processing, motor activity, or both.

To dissociate these contributions, we fit a linear mixed-effects model (*Figure 4B*) of baseline-corrected ΔF/F during the tone window (0–1 s after onset). Fixed factors were frequency and intensity, with three covariates accounting for movement- and baseline-related effects: tone-evoked head speed (0–1 s), baseline head speed (–0.5–0 s), and baseline ΔF/F. Covariates were standardized within the baseline and tone windows, so that estimated marginal means of tone *ΔF/F* are evaluated at average covariate values. Random effects were specified as sessions nested within mice (in lme4 notation: *ΔF/F ~ (Freq * Int) * Tone Speed + (Freq * Int) * Baseline Speed + (Freq * Int) * Baseline ΔF/F + (1|Subj/Ses)*).

The model revealed strong main effects of intensity ($\chi^2(1)=46.03$, *p*<0.0001) and frequency ($\chi^2(4)=66.03$, *p*<0.0001), indicating robust auditory modulation of STN activity across tones. Covariates also contributed significantly, with tone-evoked head speed ($\chi^2(1)=60.09$, *p*<0.0001), baseline head speed ($\chi^2(1)=12.23$, *p*=0.00047), and baseline ΔF/F ($\chi^2(1)=1032.24$, *p*<0.0001) all predicting tone ΔF/F, showing that both ongoing movement and pre-tone neural state strongly shaped STN activation. Significant interactions were observed between intensity and frequency ($\chi^2(4)=12.85$, *p*=0.012), intensity and baseline head speed ($\chi^2(1)=8.44$, *p*=0.0037), and frequency and baseline head speed ($\chi^2(4)=10.41$, *p*=0.034), with a trend for intensity×baseline ΔF/F ($\chi^2(1)=3.8$, *p*=0.05). These results indicate that tone-evoked STN activity depends jointly on auditory and movement properties but also on the behavioral/neural state prior to stimulus onset.

To clarify the effects, we estimated marginal means for ΔF/F in the tone window while holding movement and baseline covariates at their centered (average) values. *Figure 4* shows these baseline-corrected ΔF/F estimates (*Figure 4B*) alongside observed tone-window movement (*Figure 4C*). Higher frequency tones (8–16 kHz) exhibited significant intensity-dependent increases in STN activity (8 kHz: t(4489)=5.18, *p*<0.0001; 12 kHz: t(4479)=4.85, *p*<0.0001; 16 kHz: t(4474)=3.17, *p*=0.0015), even after accounting for tone-evoked movement and baseline effects. Moreover, the 8–12 kHz range produced the strongest STN responses, particularly at high intensity.

These results indicate that although tone-evoked movement accounted for a substantial portion of STN activation, auditory factors also made independent contributions, especially at salient high frequencies. Thus, STN responses to tones reflect a mixture of sensory and motor influences, rather than being reducible to movement alone.

## STN neurons activate during goal-directed avoidance contingencies

We then measured STN neuron activation as mice sequentially performed four well-characterized avoidance procedures (AA1-4; 7 sessions per procedure in 6 mice) signaled by auditory tones in a shuttle box (*Figure 5A*), which reliably produce distinct behavioral adaptations (e.g. *Hormigo et al., 2021a*; *Zhou et al., 2022*). *Figure 5B* shows the behavioral performance across tasks, including the percentage of active avoids (closed black circles), avoid latencies (closed orange triangles), and inter-trial crossings (ITCs, cyan bars).

In AA1, mice avoid the aversive US by shuttling between compartments (action) when CS1 was presented, performing a high percentage of correct actions (active avoids) and few errors (escapes, triggered by the US at 7 s from CS onset). In AA2, ITCs were punished by a short US, and avoid latencies reliably shifted longer in an apparent reflection of caution (*Zhou et al., 2022*). AA3 introduced a challenging discrimination, where mice continued to actively avoid in response to CS1 but were required to withhold the action during CS2 to passively avoid a short US (ITCs are no longer punished); shuttling in CS2 is a passive avoid error. In AA4, mice actively avoided in response to three

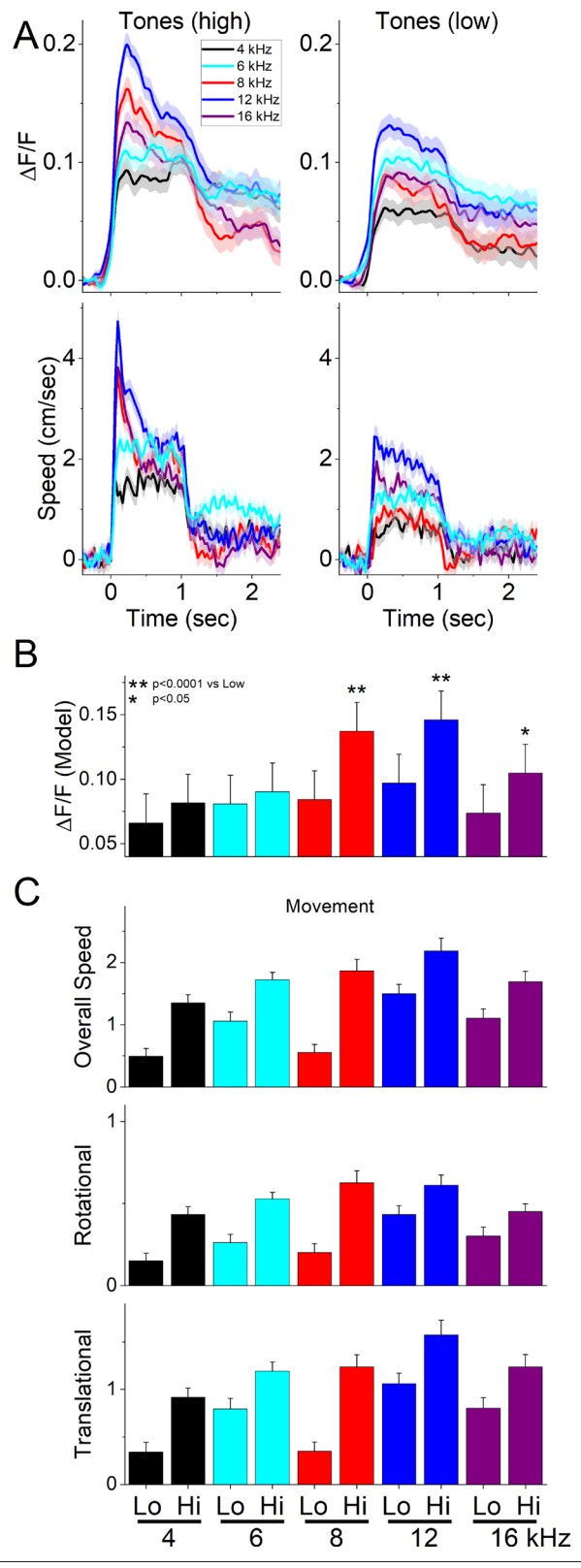

**Figure 4.** Subthalamic nucleus (STN) glutamatergic neurons discharge to auditory tones in association with movement. (**A**) Example ΔF/F calcium imaging and movement traces (mean ± SEM) evoked from STN neurons by auditory tones (1 s) of different saliency. The tones vary in frequency (4–16 kHz) and sound pressure level (SPL) (low and high dB). (**B**) Marginal means (ΔF/F) from the linear mixed-effect model for each tone. Asterisks show

*Figure 4 continued on next page*

*Figure 4 continued*

significant differences between low and high dB for the same frequency. (**C**) Overall movement and components (rotational and translational) measured during a time window (0–1 s) after tone onset corresponding to the model in B. n=6 mice.

distinct CS tones signaling different avoidance intervals (4, 7, and 15 s), producing adaptively scaled avoidance latencies.

*Figure 5C* shows ΔF/F and movement traces from CS onset for the AA1 (black), AA2 (red), and AA3 (green) avoidance procedures classified as correct responses (left panel, avoids; for AA3-CS2, correct passive avoids are shown in blue) or errors (right panel, escapes). During the avoidance procedures, the CS that drives active avoids (*Figure 5C and D* black, red, green) caused a sharp and fast (<0.5 s) ΔF/F peak at CS onset. This activation is associated with the typical orienting head movement evoked by the CS, which varies depending on task contingencies and SPL (*Zhou et al., 2023*). In contrast, AA3-CS2, which drives passive avoids, also produced a sharp STN activation at CS2 onset, even though it was associated with little orienting head movement. Thus, AA3-CS2 is distinct because it evokes an orienting STN activation without orienting movement.

To examine the effects of task contingency and response outcome on STN activation across behavioral epochs, we fit separate linear mixed-effects models for each window: baseline (−0.5–0 s pre-CS), orienting (0–0.5 s post-CS), avoid interval (0.5–7 s post-CS), and from-action (−2–2 s around the action). Fixed factors were task contingency (AA1, AA2, AA3) and outcome (correct, error). Covariates included window-specific head speed, baseline head speed (−0.5–0 s), and baseline ΔF/F (excluded in the baseline window). Covariates were standardized within each window to evaluate the estimated marginal means of ΔF/F at average covariate values. For the avoid window, the window-specific head speed covariate was standardized separately by outcome, because the movement speed differs markedly across outcomes (during the avoid interval, active avoids involve much more movement than passive avoids or escapes, which occur later after US onset). As a result, the avoid window model reflects STN activity without controlling for speed differences, whereas the from-action window model compares avoids and escapes while controlling for movement. Random effects were sessions nested within mice.

During the baseline window, STN activity differed across task contingencies and outcomes (*Figure 5E*). The linear mixed-effects model revealed main effects of contingency ($\chi^2(3)=8.23$, $p=0.042$) and baseline head speed ($\chi^2(1)=170.78$, $p<0.0001$). Critically, there was a significant contingency ×outcome interaction ($\chi^2(3)=17.13$, $p=0.00066$), indicating that the relationship between baseline activity and performance differed across tasks. Comparisons showed that in AA1–3, baseline ΔF/F did not distinguish correct and incorrect responses. In contrast, in passive avoidance trials (AA3-CS2), baseline ΔF/F was significantly higher on incorrect trials (passive avoidance errors) than on correct trials (t(236) = 4.91, $p<0.0001$). These results indicate that while baseline STN activity does not bias responding in active avoidance tasks, it strongly predicts errors in passive avoidance, with elevated baseline ΔF/F associated with unsuccessful inhibition of action. Thus, baseline STN state can undermine behavioral inhibition when the correct response is to withhold action, highlighting the need to account for baseline movement and neural activity when interpreting CS-evoked responses as done in the other windows.

In the orienting window, STN ΔF/F was jointly shaped by task contingency, response outcome, and orienting speed. The linear mixed model revealed significant main effects of contingency ($\chi^2(3)=12.90$, $p=0.0049$), orienting speed ($\chi^2(1)=29.24$, $p<0.0001$), and interactions of contingency×outcome ($\chi^2(3)=22.78$, $p<0.0001$). Baseline speed and baseline ΔF/F had no independent effect. Post-hoc contrasts (*Figure 5E*) showed that in AA3-CS1 trials, STN activation was stronger on escapes (errors) compared to correct active avoids (t(181) = 4.36, $p<0.0001$). Likewise, in AA3-CS2, STN activation was greater on incorrect actions (passive avoid errors) compared to correct passive avoids (t(183) = 2.89, $p=0.0043$). Thus, after controlling for orienting movement amplitude and baseline neural and movement activity, enhanced STN activity during the orienting epoch predicted errors in the more challenging avoidance task (AA3). This suggests that excessive orienting-related recruitment of STN may bias behaviors toward errors in challenging environments, perhaps reflecting a neural and behavioral state linked to mistaken actions.

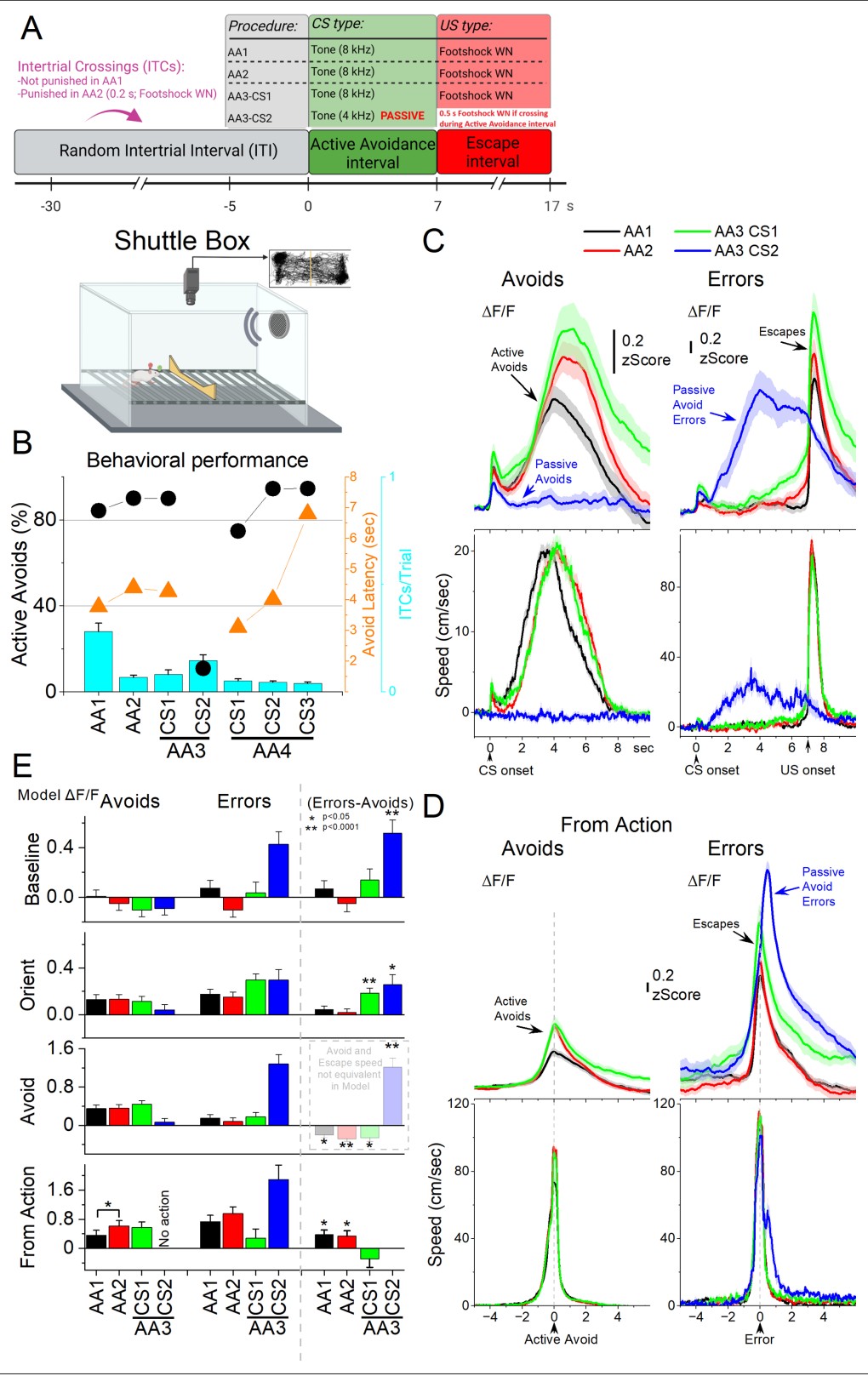

**Figure 5.** Subthalamic nucleus (STN) glutamatergic neuron activation in the context of signaled active avoidance.
(**A**) Schematic of the shuttle box and the structure of signaled avoidance trials. Each trial begins at time 0 following
a random intertrial interval (ITI). The avoidance interval lasts 7 s and is signaled by a CS (8 kHz tone); shuttling
during this window prevents the escape interval (AA1, AA2, AA3-CS1 trials). If the mouse fails to avoid, the escape

*Figure 5 continued on next page*

*Figure 5 continued*

interval begins and the US (footshock plus white noise) is delivered until escape, which occurs rapidly. AA2 is like AA1, but intertrial crossings (ITCs) are punished (0.2 s US). In AA3, CS1 continues like in AA1/2 (ITCs are no longer punished). In AA3, CS2 trials (4 kHz tone), CS signals passive avoidance: shuttling during CS2 is punished with a brief US (0.5 s). AA4 (not shown) is identical to AA1 but with variable-duration avoidance intervals signaled by three different CSs (CS1-CS3, 8, 4, 12 kHz). (**B**) Behavioral performance during the four different avoidance procedures (AA1-4) showing the percentage of active avoids (black circles), avoidance latency (orange triangles), and ITCs (cyan bars). In AA3-CS2, active avoids are passive avoid errors. (**C**) Average ΔF/F and overall movement traces aligned from CS onset for AA1, AA2, and AA3 (CS1 and CS2) procedures for trials classified as avoids (correct active or passive avoids, left) or errors (escapes or passive avoid errors, right) of CS-evoked responses. (**D**) Same as in C, but aligned from-action occurrence. (**E**) Marginal means (ΔF/F) from the linear mixed-effect models for the baseline, orienting, avoidance, and from-action windows during AA1, AA2, and AA3 (CS1 and CS2). The right panel shows estimated differences between errors and avoids, with asterisks indicating significance. Transparency in the avoidance window denotes that movement during this window was not controlled (held constant) between avoids and errors in the model. n=6 mice.

In the avoid window, STN ΔF/F was influenced by contingency, outcome, avoid speed, and both baseline covariates. The mixed-effects model revealed significant main effects of contingency ($\chi^2(3)=8.89$, $p=0.031$), outcome ($\chi^2(1)=128.82$, $p<0.0001$), avoid speed ($\chi^2(1)=37.19$, $p<0.0001$), baseline speed ($\chi^2(1)=13.94$, $p=0.00018$), and baseline ΔF/F ($\chi^2(1)=56.31$, $p<0.0001$), as well as a contingency ×outcome interaction ($\chi^2(3)=27.49$, $p<0.0001$). Post-hoc contrasts revealed no differences across active avoids or escapes among the three tasks (AA1-3), indicating that task contingency was not encoded by the population activity once the effect of movement was controlled. The contrasts also confirmed reliable differences between correct and incorrect actions in all contingencies (*Figure 5E*). However, because avoid speed was not controlled in this window (by design), the outcome-based contrasts largely reflect the ongoing movement itself. Direct comparisons of action-related STN activity are, therefore, more appropriately evaluated in the from-action window.

In the from-action window, STN ΔF/F was shaped by task contingency, response outcome, and movement covariates. The mixed-effects model revealed significant main effects of contingency ($\chi^2(3)=44.89$, $p<0.0001$), outcome ($\chi^2(1)=25.5$, $p<0.0001$), and covariates, including action speed ($\chi^2(1)=20.57$, $p<0.0001$), baseline speed ($\chi^2(1)=9.58$, $p=0.002$), and baseline ΔF/F ($\chi^2(1)=7.33$, $p=0.0067$). Post-hoc contrasts controlling for action speed and the baseline covariates revealed that STN activation during active avoids increased in AA2 compared to AA1 (t(210) = 2.78, $p=0.017$), whereas escapes did not differ across contingencies. In addition, passive avoid errors in AA3-CS2 elicited the strongest STN activation of all actions, exceeding that observed during and escapes across contingencies (AA1-3; $p<0.05$). Escapes also showed greater activation than active avoids, but only in the simpler AA1 and AA2 avoidance tasks (AA1: t(176) = 2.9 p=0.004; AA2: t(141) = 2.33, $p=0.021$), not in AA3-CS1 (*Figure 5E*). These results indicate that STN activation aligned to actions increases as animals adopt more cautious strategies in AA2, and that errors evoke stronger responses than correct actions in simpler contingencies, whereas in the more demanding AA3 condition, the strongest responses occur during passive avoid errors—highlighting the influence of behavioral context on STN coding.

Across windows, these analyses controlling for movement and baseline effects reveal that STN activation reflects a dynamic interplay between baseline state, orienting responses, and action outcomes, with distinct contributions depending on task demands. Elevated baseline ΔF/F predicted errors specifically in passive avoidance, suggesting that pre-CS state biases performance when behavioral inhibition is required. During the orienting window, excessive STN activation predicted errors in the more difficult AA3 contingencies, even after accounting for orienting movement and baseline activity. In the from-action window, where action movement and baseline covariates were controlled, STN activity tracked the development of cautious actions between AA1 and AA2 and differentiated errors by contingency: activation was stronger for escapes in the simpler AA1-2 tasks but shifted to passive avoid errors in the more challenging AA3. Together, these findings indicate that STN responses during avoidance are shaped by behavioral context and task difficulty. Baseline state biases performance before action onset, while action-aligned signals most strongly reflect the emergence of cautious responding and error encoding in a contingency-dependent manner. The pronounced activation

during errors is consistent with a role for STN in signaling escape urgency, though it may also reflect sensitivity to punishment delivered during errors.

## Heightened baseline STN activity biases urgent decisions to errors

Building on the preceding analyses of AA1–3, where STN activity distinguished correct from erroneous responses, we next examined AA4, which manipulates decision urgency by varying the duration of the avoidance interval signaled by three different CSs (*Hormigo et al., 2021a*). This design tests whether STN activity reflects not only the outcome but also the temporal urgency imposed by each CS.

Accordingly, STN activation shifted with the timing of avoidance movements (*Figure 6A*, left). When responses were aligned to the from-response window (*Figure 6A*, right), CS1 active avoids were executed at higher speed compared to CS2 (Tukey t(62) = 6.01, *p*=0.0002) and CS3 (t(62) = 8.85, *p*<0.0001), consistent with the more imminent threat signaled by CS1's shorter 4 s interval. Despite these speed differences, peak STN activation did not vary across CSs (RMANOVA F(2,62) = 1.0, *p*=0.37).

To further test how CS (urgency) and outcome shape activity, we fit separate linear mixed-effects models for the baseline, orienting, and from-action windows (*Figure 6B*). A model of the avoid window was excluded because it varies in duration per CS. Fixed factors were the CS and outcome. Covariates included window-specific head speed, baseline head speed (−0.5–0 s), and baseline ΔF/F (excluded in the baseline window). Covariates were standardized within each window to evaluate estimated marginal means of ΔF/F at average covariate values.

In the baseline window, STN activity was influenced by outcome and baseline head speed but not by CS identity (urgency level). The mixed-effects model revealed significant main effects of outcome ($\chi^2$(1)=18.24, *p*<0.0001) and baseline head speed ($\chi^2$(1)=71.50, *p*<0.0001), but no CS × outcome interaction. Post-hoc contrasts showed that in CS1 (short 4 s CS), baseline ΔF/F was significantly higher prior to errors (escapes) compared to active avoids (t(320) = 4.44, *p*<0.0001). These results indicate that heightened baseline STN activity predicts erroneous responding when decisions are made under urgent (CS1) temporal constraints, but not at longer delays. This suggests that baseline STN state can bias avoidance performance in high urgency contexts, undermining accuracy when the timing demands are extreme.

In the orienting window, STN activity was significantly influenced by outcome and covariates, with no main effect of CS. The mixed-effects model revealed a main effect of outcome ($\chi^2$(1)=12.19, *p*=0.00048) and strong effects of orienting speed ($\chi^2$(1)=23.97, *p*<0.0001), baseline head speed ($\chi^2$(1)=12.44, *p*=0.00042), and baseline ΔF/F ($\chi^2$(1)=81.55, *p*<0.0001) but not a CS × outcome interaction. Post-hoc contrasts showed that in CS3 (long 15 s CS), orienting-related STN activation was significantly stronger on escapes compared to active avoids (t(295) = 3.58, *p*=0.0004), controlling for orienting movement, baseline movement, and baseline STN activity. These results indicate that enhanced orienting-related STN activity predicts errors specifically in the long CS condition, with a trend in the same direction for the other CSs.

In the from-action window, STN activity showed robust sensitivity to outcome and CS. The mixed-effects model revealed main effects of CS ($\chi^2$(2)=15.24, *p*=0.00049) and outcome ($\chi^2$(1)=139.82, *p*<0.0001), along with a significant CS × outcome interaction ($\chi^2$(2)=13.10, *p*=0.0014). Covariate effects were also present for action speed ($\chi^2$(1)=14.20, *p*=0.00016) and baseline ΔF/F ($\chi^2$(1)=11.06, *p*=0.00088), but not for baseline head speed. Post-hoc contrasts showed that STN activation was consistently stronger for escapes compared to active avoids across all CS conditions (CS1: t(266) = 5.72, *p*<0.0001; CS2: t(267) = 7.93, *p*<0.0001; CS3: t(289) = −2.40, *p*=0.017). Together, these findings demonstrate that from-action STN responses predict trial outcomes, with stronger error-related activation across contingencies even after controlling for action movement, baseline movement, and baseline neural activity.

Across windows, the results suggest a role for STN activity in predicting errors under conditions of urgency. Elevated baseline activation biased behavior toward errors in the most demanding timing condition, while during actions, STN activity robustly differentiated outcomes, with consistently stronger activation for errors across contingencies. Although these error-related signals remained significant when controlling for movement and baseline covariates, their magnitude may reflect sensitivity to the aversive consequences of punishment.

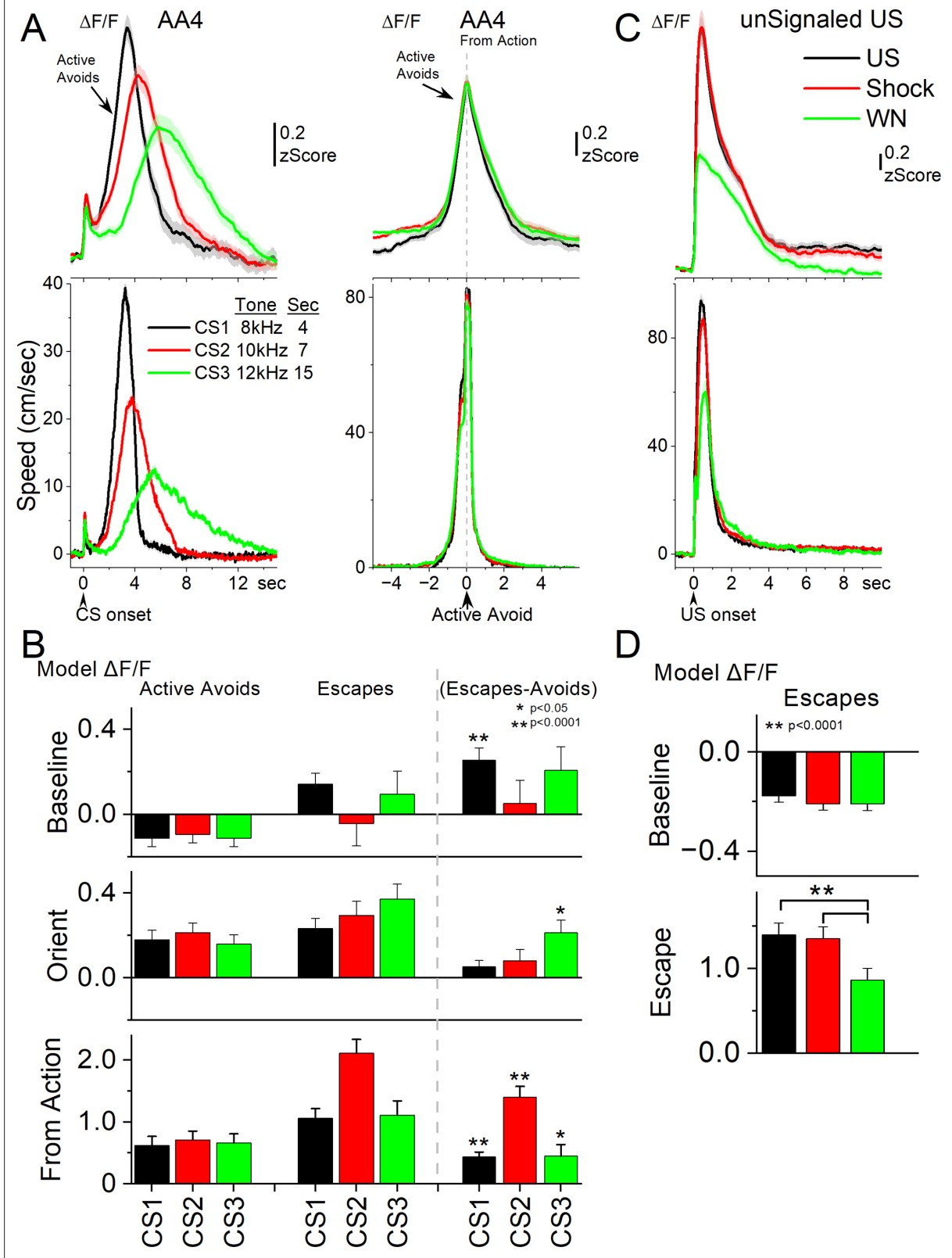

**Figure 6.** Subthalamic nucleus (STN) glutamatergic neurons track the avoidance and escape movement. (**A**) Average ΔF/F and overall movement traces from CS onset (left) and action occurrence (right) for active avoids during the AA4 procedure, which includes three CSs that signal avoidance intervals of different durations. (**B**) Marginal means (ΔF/F) from the linear mixed-effect models for the baseline, orienting, and from-action windows for the data in A. The right panel shows estimated differences between escapes. and active avoids, with asterisks indicating significance. (**C**) Average ΔF/F and overall

*Figure 6 continued on next page*

*Figure 6 continued*

movement traces from US onset for escapes during the unsignaled US procedure, which includes the US, or each of its components delivered alone (foot-shock and white noise). (**D**) Marginal means (ΔF/F) from the linear mixed-effect models for the baseline and escape windows for the data in C. n=7 mice.

## STN activation reflects painful sensation and escape vigor

In AA1-4, errors consistently produced stronger STN activation than correct avoids. Because errors are punished by the US, stronger activation could reflect the associated fast movement, the painful foot-shock, or both. To dissociate these variables, we conducted unsignaled escape sessions where the US or its components, the foot-shock or the white noise are presented alone. The full US and the foot-shock cause pain and fast movement, while white noise alone causes escape without the pain (*Figure 6C*; 7 mice). The unsignaled US reliably evoked strong STN activation in association with fast escapes. Foot shock alone produced STN activation and escape behavior comparable to the full US, whereas white noise alone drove slower escapes with weaker STN activation (speed: Tukey t(10) = 4.4, p=0.02; ΔF/F: Tukey t(10) = 4.96, p=0.001 vs foot-shock).

We evaluated these effects by fitting separate linear mixed-effects models (*Figure 6D*) for the baseline and escape windows (0–4 s after US onset). The sole fixed factor was the US with three levels (foot-shock, white noise, or both). Covariates included window-specific head speed, baseline head speed, and baseline ΔF/F (excluded in the baseline window). Covariates were standardized within each window to evaluate estimated marginal means of ΔF/F at average covariate values.

In the baseline window, linear mixed-effects modeling revealed no main effect of US condition ($\chi^2(2)$=1.30, p=0.52). In contrast, baseline head speed was a strong predictor of STN activity ($\chi^2(1)$=60.76, p<0.0001). Post-hoc contrasts confirmed that baseline ΔF/F did not differ across foot-shock, foot-shock +white noise, or white noise alone conditions (all p>0.99). Thus, before stimulus onset, STN activity primarily tracked ongoing movement rather than anticipating the sensory or aversive properties of the upcoming US, which is unpredictable.

In the escape window, STN activity showed strong modulation by US condition ($\chi^2(2)$=130.54, p<0.0001) and was also independently predicted by baseline ΔF/F ($\chi^2(1)$=40.69, p<0.0001) and by escape vigor (head speed; $\chi^2(1)$=8.52, p=0.0035). Post-hoc contrasts revealed that both the full US and foot-shock alone produced significantly stronger ΔF/F responses compared to white noise (shock vs. WN: t(81) = 10.37, p<0.0001; shock +WN vs. WN: t(86) = 8.46, p<0.0001), whereas responses to the full US and foot-shock alone did not differ (t(78) = 0.79, p=0.43). Thus, during escape behavior, STN activation scaled both with the painful foot-shock and the vigor of the resulting movement, with white noise driving weaker responses consistent with its lower behavioral impact.

Since the models controlled for movement and baseline covariates, the stronger activation during foot-shock compared to white noise indicates a specific contribution of painful sensation. This may also account for the stronger STN activation observed during active and passive avoid errors in AA1–4. Overall, these results show that STN activity scales with escape vigor and is further enhanced by nociceptive input, even when movement is controlled. Although urgency is maximal in the presence of the US—where painful sensation and urgency are tightly intertwined—there was little evidence from the CS1–3 avoidance trials in AA4 that urgency alone predicted STN activity independently of movement vigor. Thus, STN coding during punishment appears to integrate painful sensory signals with motor output, providing a mechanism by which aversive stimuli and the urgency-linked vigor of escape jointly shaped subthalamic responses.

## STN neurons exhibit distinct temporal relationships with movement

The preceding results during avoidance and escape tasks measured population STN neuron activation. To investigate single-neuron activity in the STN, we imaged calcium dynamics with miniscopes during signaled active avoidance. We first quantified each neuron's relation to movement by computing the cross-correlation between speed and ΔF/F. K-means clustering of these cross-correlation functions revealed four classes of neurons (797 neurons, 5 mice; *Figure 7A*). Class 1 (66.6%) showed no relation to speed, and Class 2 (17%) had significant but weak correlations. The remaining neurons (16.4%) exhibited strong correlations but differed in their temporal lag: Class 3 (9.4%) showed rightward lag relative to movement, while Class 4 (7%) showed leftward lag. When plotted during active avoidance, Class 4 neurons activated prior to movement onset (*Figure 7B and C*, red), suggesting a role in

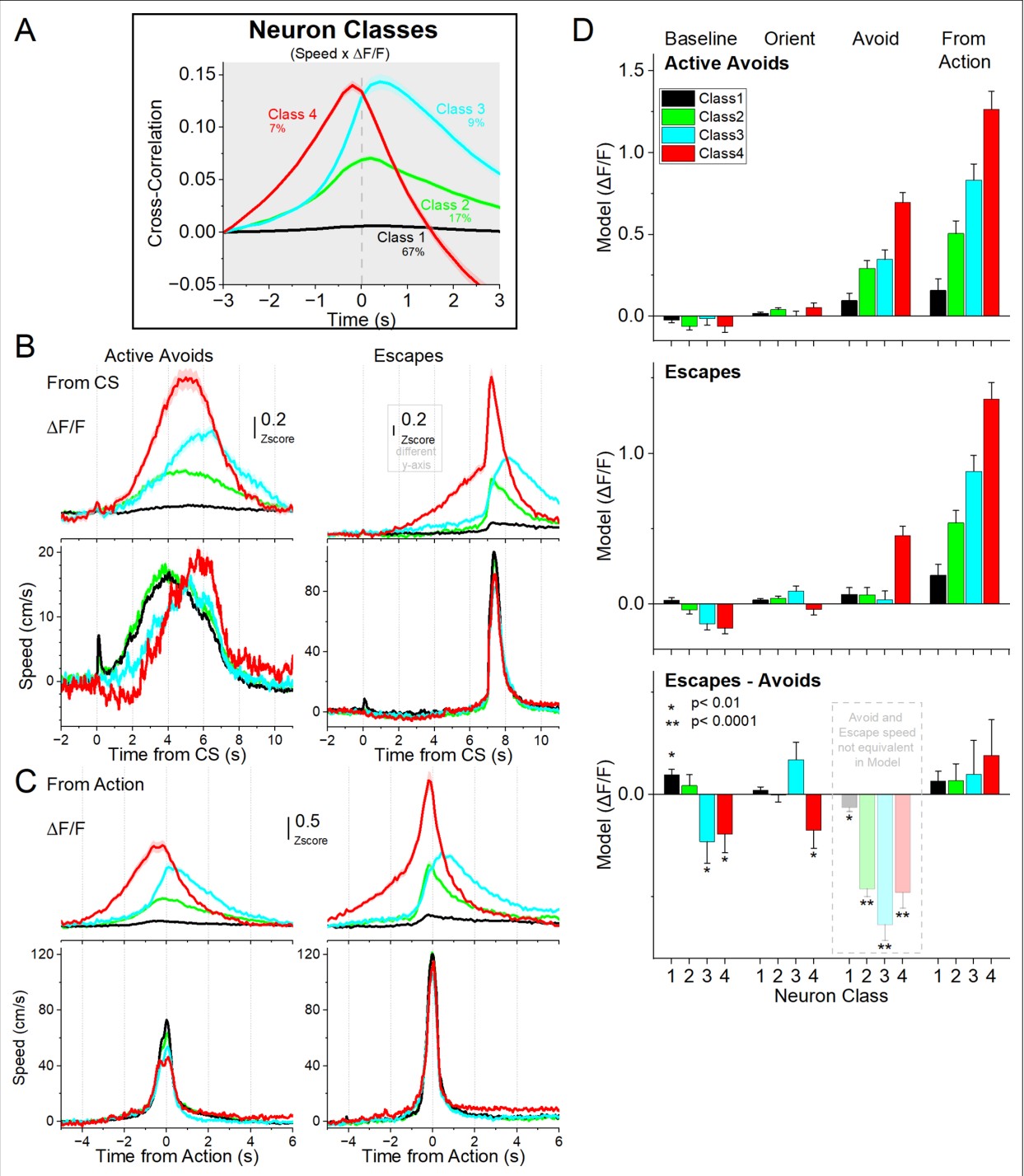

**Figure 7.** Different classes of subthalamic nucleus (STN) glutamatergic neurons during signaled active avoidance, classified by the cross-correlation between head speed and ΔF/F. (**A**) k-means clustering of the cross-correlation time series identified four distinct classes of neurons. Class 1 neurons showed little cross-correlation with movement. Class 2 showed moderate correlation around zero lag. Class 3 and Class 4 exhibited stronger cross-correlations, with Class 4 activity preceding and Class 3 following the head movement zero lag. (**B**) Average ΔF/F and overall movement traces aligned from CS onset for active avoids (left) and escapes (right) during active avoidance procedures (AA1-3 CS1 combined). (**C**) Same as in B, but aligned from-action occurrence. (**D**) Marginal means (ΔF/F) from the linear mixed-effect models for the baseline, orienting, avoidance, and from-action windows during AA1, AA2, and AA3-CS1 shown separately for active avoids (top) and escapes (middle). The bottom panel shows estimated differences between escapes and active avoids, with asterisks indicating significance. Transparency in the avoidance window indicates that movement during this window was not controlled (held constant) between active avoids and escapes in the model. n=7 mice.

preparing or initiating avoidance, whereas Class 3 neurons activated during the movement (**Figure 7B and C**, cyan), consistent with maintaining or monitoring ongoing actions.

To avoid confusion between analyses, it is important to note that the movement-sensitivity classes defined here (Class 1–4; **Figure 7**) are conceptually distinct from both the movement-onset classes (Class A-C; **Figure 3**) and the neuronal activation 'types' introduced later in the avoidance-mode analysis. The Class 1–4 grouping reflects how neurons relate to movement across the entire session, based on their cross-correlation with speed. The onset classes A–C capture neural activity specifically around spontaneous movement initiation during general exploration. In contrast, the later activation 'types' are derived within each avoidance mode and describe how neurons differ in their activation patterns during identical CS1 avoidance responses. These classifications answer different questions about STN function and are not intended to correspond to one another. In addition, it is worth emphasizing that the entire STN population is not recorded within a single session; each session contributes only a subset of neurons to the dataset. Consequently, each neural class is composed of neurons drawn from partially non-overlapping sets of sessions, each with its own movement traces. For this reason, we plot movement traces separately for each neural class to maintain strict within-session correspondence between neural activity and the behavior collected in the same sessions.

We next asked whether neuron movement-sensitivity classes differed in their modulation during active avoidance by fitting mixed-effects models (**Figure 7D**) of $\Delta F/F$ on CS1 active avoidance trials across tasks (AA1–3) for each window: baseline (–0.5–0 s pre-CS), orienting (0–0.5 s post-CS), avoid interval (0.5–7 s post-CS), and from-action (–3.5–3 s around the action). Fixed factors included task (AA1–3), CS intensity (dB), outcome (avoid vs escape), and neuron class (from the cross-correlation clustering). However, we focused the analysis on the outcome and neuron class factors, as different neurons were recorded across sessions. As with the photometry, covariates included window-specific head speed, baseline head speed (–0.5–0 s), and baseline $\Delta F/F$ (excluded in the baseline window). Covariates were standardized within each window to evaluate the estimated marginal means of $\Delta F/F$ at average covariate values. For the avoid window, the window-specific head speed covariate was standardized separately by outcome. Neurons were nested within mice as random effects.

In the baseline window preceding the CS, STN neuron classes exhibited distinct outcome-related modulation (**Figure 7D**). Mixed-effects modeling revealed significant main effects of outcome ($\chi^2(1)=5.71$, $p=0.016$), neuron class ($\chi^2(3)=42.19$, $p<0.0001$), and baseline head speed ($\chi^2(3)=107.03$, $p<0.0001$), as well as an outcome × class interaction ($\chi^2(3)=14.17$, $p=0.0027$). Post-hoc contrasts showed that Class 1 neurons, which lacked a clear correlation with movement, displayed elevated baseline activity on escapes relative to active avoids ($t(3537)=3.45$, $p=0.00056$). By contrast, Class 3 and 4 neurons, which correlated strongly with movement, showed stronger baseline activation during active avoids compared to escapes (Class 3: $t(3585)=2.17$, $p=0.029$; Class 4: $t(3584)=2.09$, $p=0.035$). These results indicate that baseline STN activity encodes upcoming trial outcome in a class-specific manner, with movement-linked neurons preferentially recruited before successful trials and movement-independent neurons more active before errors. This suggests that distinct subpopulations bias performance even before cue onset.

In the orienting window, mixed-effects modeling revealed no significant main effects of outcome or class, and no outcome × class interaction. In contrast, baseline head speed ($\chi^2(1)=17.05$, $p<0.0001$) and baseline $\Delta F/F$ ($\chi^2(1)=485.21$, $p<0.0001$) had strong main effects, emphasizing the importance of controlling their influence. Comparisons showed only a weak trend for Class 4 neurons to activate more strongly during active avoids than escapes ($t(2992)=2.02$, $p=0.043$). Thus, STN activity during the orienting phase was dominated by baseline state and movement covariates, with only marginal evidence for class- or outcome-specific modulation.

In the avoidance interval window, STN neuron classes exhibited robust outcome-related modulation, though outcome effects were not controlled by avoid speed (**Figure 7D**). Mixed-effects modeling revealed significant main effects of outcome ($\chi^2(1)=364.79$, $p<0.0001$; uncontrolled for speed), neuron class ($\chi^2(3)=393.06$, $p<0.0001$), and their interaction ($\chi^2(3)=349.01$, $p<0.0001$). Covariates also had strong influences, including avoidance speed ($\chi^2(1)=29.21$, $p<0.0001$), baseline head speed ($\chi^2(1)=18.11$, $p<0.0001$), and baseline $\Delta F/F$ ($\chi^2(1)=1518.07$, $p<0.0001$). Post-hoc contrasts focused within each outcome revealed that during active avoids, all neuron classes significantly differed from one another ($p<0.0001$), except Class 2 and 3 ($p=0.2$). For escapes, Class 4 neurons differed from all other classes ($p<0.0001$), which did not differ among themselves.

In the from-action window, STN ΔF/F was shaped by both outcome and class. The mixed-effects model revealed significant main effects of outcome ($\chi^2(1)$=19.55, $p$<0.0001), neuron class ($\chi^2(3)$=596.55, $p$<0.0001), and their interaction ($\chi^2(3)$=21.46, $p$=0.0057). Covariates again had strong influences, including action window speed ($\chi^2(1)$=34.44, $p$<0.0001), baseline speed ($\chi^2(1)$=10.01, $p$=0.0015), and baseline ΔF/F ($\chi^2(1)$=239.47, $p$<0.0001). Post-hoc contrasts showed that none of the neuron classes differed between active avoids and escapes after controlling for covariates, indicating that they did not differentially encode errors. However, all four neuron classes differed significantly from one another within both avoids and escapes ($p$<0.0001). These results indicate that STN neurons classified based on their relationship to movement were distinctly engaged around the time of action but did not encode errors.

Across windows, these results demonstrate that STN activity is dynamically structured by time, task demands, and neuron class. Before cue onset, baseline state biases upcoming performance in a class-specific fashion. During orientation, activity largely reflects baseline and movement factors, with little outcome specificity. Finally, around the action, class differences are robust without encoding errors. Together, these findings indicate that STN coding of avoidance emerges through the interplay of baseline bias and movement-related drive, with distinct classes providing complementary contributions to predicting and executing avoidance under threat.

An additional observation was that Class 4 neurons, which showed activation preceding the avoid actions, tended to be associated with late avoids from CS onset, which often reflects a cautious response strategy. This was evident in *Figure 7B* speed traces (aligned from CS, red trace). Indeed, the time-to-peak speed for avoid actions was significantly longer for Class 4 neurons compared to Class 1 (Tukey q(645) = 5.6, $p$=0.002) and Class 2 neurons (Tukey q(645) = 4.3, $p$=0.04). These findings suggest that STN neurons may recruit more strongly during cautious, delayed action initiation, which we examined next.

## STN neurons encode more robustly cautious actions

Whereas the previous analysis sorted STN neurons based on their relationship to movement, we next asked whether STN activity differentially encodes distinct types of active avoidance actions. To capture action variability, we applied k-means clustering directly to the time series of movement speed aligned to CS onset across active avoidance trials. This revealed three distinct modes of response (*Figure 8A*, bottom gray panel). Mode 1 responses (black) were rapid avoids initiated immediately after the orienting movement, whereas Mode 2 (red) and Mode 3 (cyan) responses were initiated later, reflecting more cautious behavior (*Zhou et al., 2022*).

Mode 2 and Mode 3 avoidance responses differed from each other and from Mode 1 in several behavioral features. First, baseline movement prior to CS onset was lowest in Mode 3 avoids compared to both Mode 2 (Tukey q=5.85, $p$=0.0001) and Mode 1 (Tukey q=18.12, $p$<0.0001), with Mode 2 exhibiting the highest spontaneous movement. Second, the amplitude of the CS-evoked orienting response was significantly reduced in Mode 3 avoids relative to Mode 2 (Tukey q=9.77, $p$<0.001) and Mode 1 (Tukey q=20.85, $p$<0.0001). Finally, the change in speed during the avoidance interval—measured as the baseline-corrected area under the speed curve—was significantly greater in both Mode 1 (Tukey q=13.38, $p$<0.0001) and Mode 3 (Tukey q=12.12, $p$<0.0001) responses compared to Mode 2, regardless of whether it was measured from CS onset or from the avoidance response.

These results indicate that both Mode 2 and Mode 3 avoid reflect cautious responding, but with distinct behavioral signatures suggestive of different internal states. Mode 2 avoids involving ongoing spontaneous movement at CS onset, accompanied by larger orienting responses and a smaller change in speed during avoidance—possibly indicating a distracted but cautious animal already in motion. In contrast, Mode 3 avoids are marked by behavioral quiescence at CS onset, minimal orienting, and delayed avoidance, consistent with an alert yet hesitating state in which the animal delays responding until the last moment. Mode 1 avoids, by comparison, show rapid onset and large-amplitude orienting responses, typical of animals adapting to dynamic or challenging environments (*Zhou et al., 2023*).

We next examined how STN activity varied across avoidance modes by averaging the ΔF/F time series from all recorded neurons. This revealed distinct levels of STN activation across the modes (*Figure 8A*, top panel, aligned to CS onset; *Figure 8B*, aligned to avoidance response). Mode 3 avoids exhibited the highest STN activation, measured both from CS onset (Tukey q=19.26, $p$<0.0001 vs. Mode 1; Tukey q=16.52, $p$<0.0001 vs. Mode 2) and from avoidance response

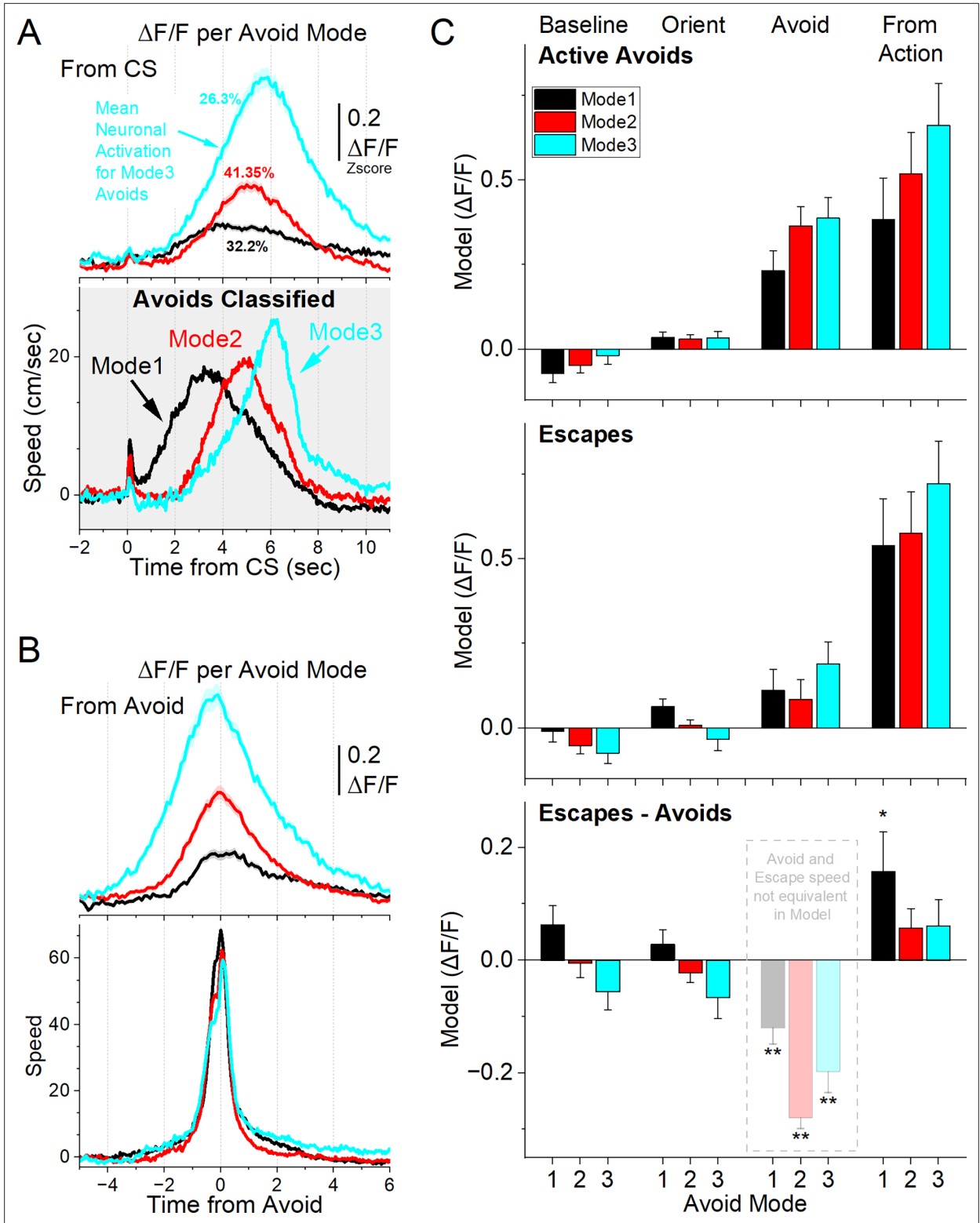

**Figure 8.** Activation of subthalamic nucleus (STN) glutamatergic neurons across distinct modes of signaled active avoidance. (**A**) k-means clustering of movement speed time series from CS onset (gray panel) revealed three distinct avoidance modes. Mode 1 avoids were initiated rapidly after CS onset, whereas Mode 2 and Mode 3 avoids were delayed, reflecting increasingly cautious responding. The top panel shows the average ΔF/F activity of all recorded STN neurons for each avoidance mode. STN activation was weak during Mode 1, intermediate during Mode 2, and strongest during Mode 3. (**B**) Same as in A, but aligned from-action (avoid) occurrence. (**C**) Marginal means (ΔF/F) from the linear mixed-effect models for the baseline, orienting, avoidance, and from-action windows during AA1-3 (CS1), shown separately for active avoids (top) and escapes (middle). The bottom panel shows

*Figure 8 continued on next page*

*Figure 8 continued*

estimated differences between escapes and active avoids, with asterisks indicating significance. Transparency in the avoidance window indicates that movement during this window was not controlled (held constant) between active avoids and escapes in the model. n=7 mice.

(Tukey q=20.95, $p<0.0001$ vs. Mode 1; Tukey q=15.68, $p<0.0001$ vs. Mode 2). In contrast, Mode 1 avoided showing the lowest STN activation. Importantly, there were no significant differences in STN activity between modes during the CS-evoked orienting response (one-way ANOVA, $F(2,2185)=0.36$, $p=0.6943$), indicating that these neurons do not encode orienting per se. Together, these results indicate that STN activation scales with behavioral caution, with the highest activation associated with the most delayed (Mode 3) avoidance responses. Since the previous analysis may reflect the effects of movement on STN activity, we assessed the results with mixed-effects models (*Figure 8C*) controlling for movement and baseline covariates in four windows as described in the previous sections. In the baseline window, STN ΔF/F prior to CS onset was strongly influenced by baseline head speed ($\chi^2(1)=79.32$, $p<0.0001$), with weaker contributions from outcome ($\chi^2(1)=4.42$, $p=0.036$), but not avoidance mode. No outcome × avoidance mode interaction was observed. Thus, baseline STN activity reflects ongoing movement rather than differences in avoidance strategy.

During the CS-evoked orienting period, STN activity was dominated by baseline covariates, including baseline STN activation ($\chi^2(1)=480.43$, $p<0.0001$) and baseline head speed ($\chi^2(1)=6.41$, $p=0.011$). There were no main effects of avoidance mode or outcome, and only a weak interaction ($\chi^2(2)=7.43$, $p=0.024$). These results suggest that STN responses during orienting largely track pre-existing activity levels rather than encoding avoidance-specific signals.

In the avoid interval window, STN activity was strongly modulated by outcome ($\chi^2(1)=260.55$, $p<0.0001$), but this was not controlled for avoid speed by design and, therefore, it reflects the large difference in movement between active avoids and escapes during the avoid epoch. In addition, there were effects of avoidance mode ($\chi^2(2)=11.31$, $p=0.0035$) and outcome x avoidance mode interaction ($\chi^2(2)=15.06$, $p=0.00053$), alongside robust effects of baseline STN activity ($\chi^2(1)=1353.90$, $p<0.0001$). Contrasts within the active avoid outcome confirmed stronger STN responses for cautious Mode 2 and Mode 3 avoids relative to rapid Mode 1 avoids Mode 1 vs Mode 2: (t(3376)=5.33, $p<0.0001$); Mode 1 vs Mode 3: (t(3264)=4.62, $p<0.0001$). Within escapes, Mode 3 produced stronger activation than Mode 2 (t(2710)=3.12, $p=0.0054$). These findings demonstrate that even after accounting for motor covariates within action outcomes, STN activity differentiates avoidance modes and preferentially encodes cautious avoidance responding represented by Mode 2 and especially Mode 3.

In the from-action window, which controls for action covariates, STN activity was strongly influenced by avoidance mode ($\chi^2(2)=27.45$, $p<0.0001$) and outcome ($\chi^2(1)=34.17$, $p<0.0001$), but no interaction between them. The covariates had strong contributions, including action speed ($\chi^2(1)=8.34$, $p=0.0038$), baseline speed ($\chi^2(1)=4.69$, $p=0.03$) and baseline ($\chi^2(1)=222.76$, $p<0.0001$) STN activity. Contrasts showed weak differences between active avoids and escapes after controlling for action speed, with only the non-cautious Mode 1 (t(2645)=2.21, $p=0.026$) showing some effect of errors. Within the active avoids, there were significant differences between the three avoidance modes, with Mode 3 producing the strongest action-related activation (Mode 1 vs Mode 2: t(3329)=3.12, $p=0.0018$; Mode 1 vs Mode 3: t(3309)=5.61, $p<0.0001$; Mode 2 vs Mode 3: t(2768)=3.38, $p=0.0014$). Within escapes, there were only weak differences between Mode 2 and Mode 3 (t(2759)=3.2, $p=0.0041$). These results indicate that STN neurons encode the avoidance mode reflecting cautious behavior, after controlling for movement and baseline covariates, with the largest responses tied to the most cautious Mode 3.

In summary, these results establish a clear relationship between STN activation and behavioral caution: across avoidance modes, the most delayed responses (Mode 3) consistently evoked the strongest STN activity, and this effect persisted even after controlling for movement and baseline covariates. Thus, STN activity does not merely reflect motor output but tracks the degree of cautious responding. However, the averaging approach used here treats STN as a homogeneous population. In the next section, we turn to analyses of neuronal subpopulations to determine whether distinct groups of STN neurons contribute differently to encoding cautious behavior.

## Subpopulations of STN neurons encode caution differently

The preceding analysis combined neuronal activity within each avoidance mode, but this could mask functional diversity among subpopulations. To address this, we applied k-means clustering to the ΔF/F time series of individual neurons, which revealed three distinct activation types during each avoidance mode (*Figure 9A and B*). Importantly, neuronal activation types are defined independently within each mode and are not assumed to overlap across modes. Thus, comparisons between modes do not reflect comparisons of the same neurons. In Mode 1 avoids, *Type 1* a (gray) neurons showed no activation, *Type 1b* (orange) neurons were inhibited at CS onset before activating during the avoidance response, and *Type 1* c (green) neurons activated during the avoidance response (Tukey q=14.88, *p*<0.0001, 1 c vs. 1 a; q=8.4, *p*<0.0001, 1 c vs. 1b). In Mode 2 avoids, *Type 2* a neurons showed no activation, *Type 2b* neurons activated during avoidance responses, and Type 2 c neurons activated in advance of the response (Tukey q=25.39, *p*<0.0001, 2b vs. 2 a; q=37.55, *p*<0.0001, 2 c vs. 2 a; q=16.96, *p*<0.0001, 2 c vs. 2b). Mode 3 avoids exhibiting similar patterns to Mode 2. *Type 3* a neurons showed minimal activation, *Type 3b* neurons activated near the time of movement, and *Type 3* c neurons activated prior to the avoidance response (Tukey q=33.49, *p*<0.0001, 3b vs. 3 a; q=41.43, *p*<0.0001, 3 c vs. 3 a; q=18.77, *p*<0.0001, 3 c vs. 3b).

Within each avoidance mode, avoidance movements were generally similar regardless of neuron type, with a few exceptions involving *Type a* neurons, which did not activate during avoidance. In Mode 1, *Type 1* a neurons were associated with higher baseline movement and stronger CS-evoked orienting responses, but lower avoidance movement speeds (from CS onset or from avoidance response) compared to *Type 1b* (Tukey q=10.55, *p*<0.0001) and *Type 1* c (Tukey q=12.25, *p*<0.0001) neuron types (*Figure 9A and B*). In Mode 2, *Type 2* a neurons were linked to larger orienting responses than *Type 2b* (Tukey q=5.36, *p*=0.0047) and *Type 2* c (Tukey q=6.67, *p*=0.0001) neurons. Together, these results suggest that stronger orienting responses predict reduced STN activation during the ensuing avoidance behavior, although this effect may be largely attributable to movement and baseline covariates.

To control for movement and baseline covariates, we next incorporated neuronal activation *Type* into the mixed-effects models described in the previous section (*Figure 9C and D*). Neuron type significantly predicted STN activity in the baseline ($\chi^2$(2)=39.48, *p*<0.0001), avoid ($\chi^2$(2)=355.33, *p*<0.0001) and from-action ($\chi^2$(2)=280.02, *p*<0.0001) windows, but not in the orienting ($\chi^2$(2)=2.04, *p*=0.36) window. Examining the outcome × avoidance mode × neuron type interaction, significant effects emerged in the baseline ($\chi^2$(4)=28.91, *p*<0.0001), avoid ($\chi^2$(4)=355.31, *p*<0.0001), and from-action ($\chi^2$(4)=16.06, *p*=0.0029) windows.

Baseline STN activity of *Type c* neurons revealed an intriguing pattern: activation was higher prior to active avoids than escapes in the more cautious modes (Mode 2: t(3424)=2.02, *p*=0.042 and Mode 3: t(3423)=3.14, *p*=0.0017), but activity was elevated in anticipation of escapes in Mode 1 (t(3425)=3.95, *p*<0.0001).

In the orienting window, neuron type did not show a main effect or three-way interaction, and contrasts revealed only a modest difference for *Type 3* a neurons, which were more active during escapes than active avoids (t(2774)=2.89, *p*=0.0038).

In the avoidance window, the STN activity of *Type b* and *Type c* neurons within active avoids differed between the cautious modes 2 and 3 (e.g. *Type 2* c vs 1 c: t(3338)=5.24, *p*<0.0001; *Type 3* c vs 1 c: t(3087)=3.52, *p*=0.00087) compared to Mode 1. Within escapes in the avoidance window, only *Type c* neurons diverged between avoidance modes, with *Type 3* c neurons activating more strongly than *Type 1* c (t(3014)=2.90, *p*=0.011) and *Type 2* c (t(2650)=2.53, *p*=0.023) neurons. This effect likely reflects late actions that failed to reach the exit before US delivery.

In the from-action window, only the least active *Type a* neurons showed higher activity for escapes than active avoids in Mode 1 and Mode 3 (*Type 1* a: t(2769)=3.97, *p*<0.0001, and *Type 3 a:* t(2595)=4.5, *p*<0.0001). By contrast, the more active neuron types did not differ by outcome but scaled robustly with caution mode within active avoids (not escapes): *Type b* neurons differed between all three modes (*Type 2b vs 3b*: t(2891)=3.49, *p*=0.00096; *Type 3b vs 1b*: t(3307)=4.38, *p*<0.0001; *Type 2b vs 1b*: t(3248)=2.32, *p*=0.02), and *Type c* neurons were more active in the most cautious Mode 3 compared to Mode 1 (*Type 1* c vs 3 c: t(3029)=3.93, *p*=0.00025), showing a trend between the other modes (*p*=0.06). These results show that STN encoding of avoidance behavior is concentrated in specific subtypes. *Types b* and *c* neurons tracked the graded expression of caution across modes, whereas *Type a* neurons remained largely insensitive to caution but encoded errors.

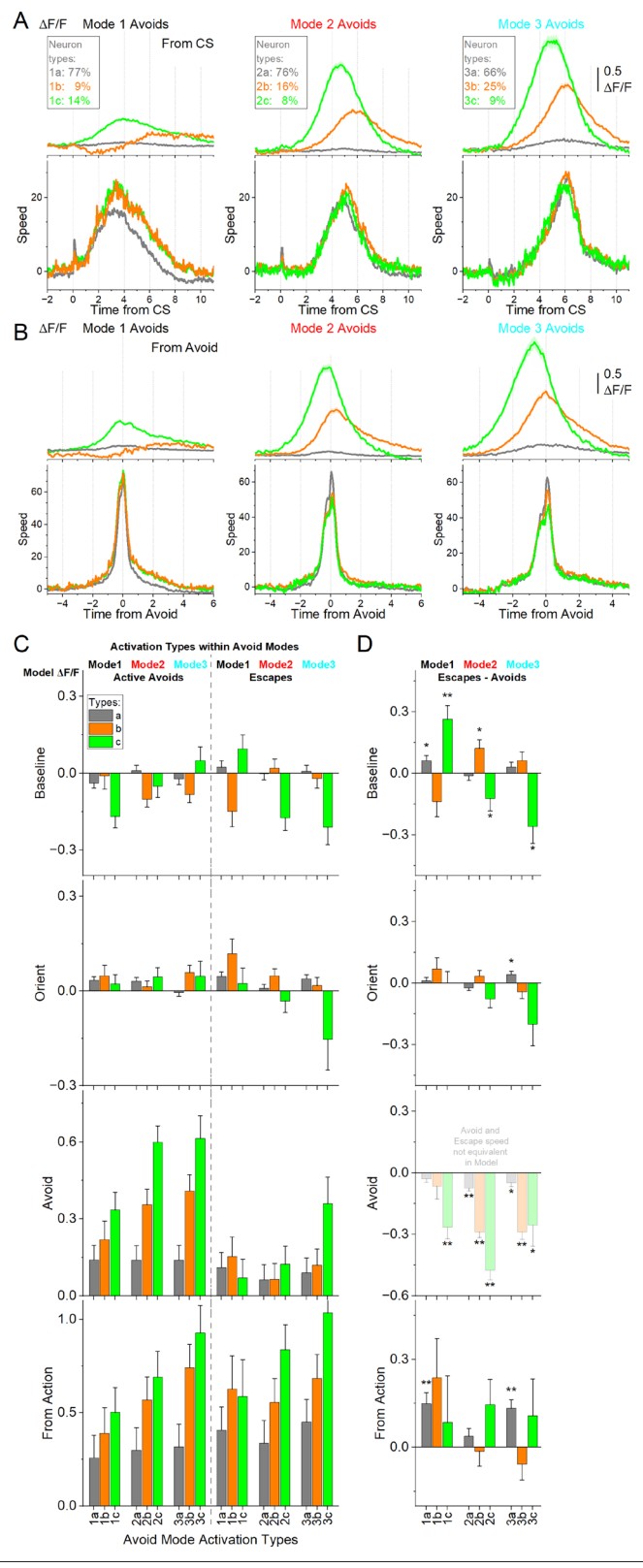

**Figure 9.** Subtypes of subthalamic nucleus (STN) glutamatergic neurons across distinct modes of signaled active avoidance. (**A**) k-means clustering of the ΔF/F time series within each avoid mode (from **Figure 8**) revealed three neuronal subtypes (**a–c**) of neurons. Type a (1–3 a) neurons showed little STN activation across modes. Type b (1-3b) neurons were activated during all avoidance modes but showed inhibition at CS onset prior to Mode 1

*Figure 9 continued*

avoids. Type c (1–3 c) neurons displayed the strongest activation overall, peaking most sharply during Mode 3 avoids, which had the longest response delays. Traces are aligned from CS onset. (**B**) Same data as in A, aligned from-action occurrence. (**C**) Marginal means (ΔF/F) from the linear mixed-effect models for the baseline, orienting, avoidance, and from-action windows during AA1-3 (CS1), shown separately for active avoids (left) and escapes (right). (**D**) Estimated differences between escapes and active avoids, with asterisks indicating significance. Transparency in the avoidance window indicates that movement during this window was not controlled (held constant) between active avoids and escapes in the model. n=7 mice.

Together, these findings demonstrate that STN coding of cautious behavior is not uniform across the population but instead emerges from the coordinated activity of distinct neuronal subtypes. *Type a* neurons contributed little to avoidance encoding but discharged during shock-driven escapes, reflecting aversive outcomes, whereas *Type b* and *Type c* neurons scaled their responses with increasing caution. Among them, *Type c* neurons showed anticipatory activation that preceded avoidance, suggesting a subset of STN neurons may actively shape action planning under threat rather than simply reflect movement execution. By integrating nociceptive signals (*Type a* neurons), motor correlates (captured by movement covariates which strongly predict STN activity), and cognitive signals related to cautious responding (*Type b* and especially *Type c neurons*), the STN functions as a heterogeneous hub where diverse information streams converge to guide adaptive avoidance behavior under uncertainty and threat.

## STN inhibition blocks signaled active avoidance

The preceding STN excitation experiments suggest that STN activation may have an important role in mediating signaled active avoidance. If this is the case, signaled avoidance should be impaired by optogenetically inhibiting STN. To inhibit glutamatergic STN neurons, we expressed eArch3.0 in the STN of Vglut2-Cre mice with bilateral injections of a Cre-inducible AAV (STN-Arch, n=6 mice).

We found that inhibiting STN neurons in CS+Light trials suppressed the percentage of active avoids compared to CS trials as a function of green light power during AA1, AA2, and AA3-CS1 (*Figure 10A*). Active avoids were suppressed by all powers tested; at the higher light powers (≥15 mW), avoids were strongly suppressed (Tukey t(5) = 11.65 *p*=0.0004 CS vs CS+Light). When avoids failed, STN inhibition did not suppress the occurrence of escape responses evoked by the US; the mice escaped rapidly at US onset on every failed avoid trial (7 s from CS onset). Thus, STN inhibition selectively suppressed avoidance responses, not escape responses. In AA1, some mice increased the number of ITCs during the intertrial interval following a failed avoid caused by STN inhibition (*Figure 10A* bottom panel, Tukey t(5) = 10.28 *p*=0.0008), which is a common coping strategy when ITCs are not punished. We also tested if STN inhibition impairs passive avoids in AA3 and found that the number of errors in CS2+Light trials was not different from that in control CS2 trials (*Figure 10A*; Tukey t(9)=0.83 *p*=0.9).

Measurements of peak speed (baseline corrected) during AA1, AA2, and AA3-CS1 showed that STN inhibition slightly increased the orienting response (*Figure 10B and C*; Tukey t(5) = 5.76 *p*=0.0096) but this was followed by a suppression of action speed (*Figure 10B and C*; Tukey t(5) = 5.06 *p*=0.016) in association with the abolishment of avoids. However, mice escaped rapidly upon US onset. Neither the peak speed (Tukey t(5) = 0.6 *p*=0.6) nor the time-to-peak speed (Tukey t(5) = 0.9 *p*=0.5) of escapes was affected by STN inhibition, indicating that STN inhibition did not paralyze the mice or impair their ability to cross. The movement during the passive avoidance interval in AA3-CS2 trials was somewhat inhibited by STN inhibition (Tukey t(4)=4.67 *p*=0.04), which would facilitate passive avoidance.

These results indicate that STN glutamatergic neuron activation is essential for signaled active avoidance.

## STN lesions impair active avoidance but not passive avoidance actions

The optogenetic experiments indicate that STN is essential for cued active avoidance actions. We decided to use lesions of the STN to verify these results and to test additional issues. First, we determined if the lesion impaired active and passive avoidance learning. Second, we determined if the lesion impaired performance of unsignaled passive avoidance (when ITCs are punished in AA2).

To determine if STN neurons are important for learning avoidance behaviors, we bilaterally injected AAV8-EF1a-mCherry-flex-dtA into the STN of Vglut2-cre mice (n=12) to lesion STN glutamatergic

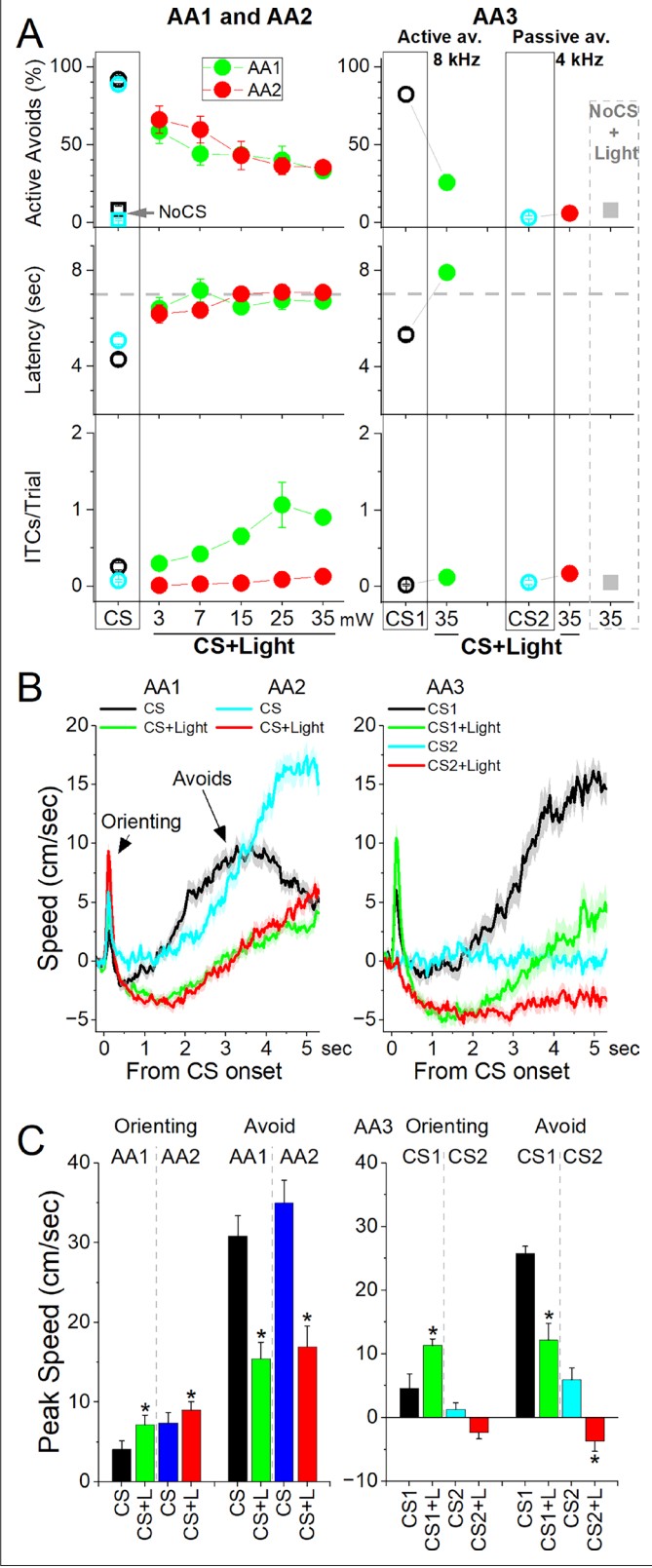

**Figure 10.** Optogenetic inhibition of subthalamic nucleus (STN) glutamatergic neurons impairs signaled avoidance. (**A**) Effect of Cont green light delivered at different powers on AA1 (green circles), AA2 (red circles), and AA3 (right panel) in mice expressing eArch3.0 in STN glutamatergic neurons. Note the strong abolishment of active avoidance responses in CS+Light trials for AA1, AA2, and AA3-CS1. In contrast, passive avoids during

*Figure 10 continued on next page*

*Figure 10 continued*

AA3-CS2 were not impaired. (**B**) Traces of overall movement (speed) during AA1, AA2, and AA3 for CS trials and CS+Light trials combined for different light powers. The trials are aligned by CS onset, which reveals the orienting response evoked by the CS followed by the ensuing avoid action. (**C**) Population data of peak speed from CS onset for orienting and avoidance responses during AA1, AA2, and AA3. Asterisks denote significant differences ($p<0.05$) between CS vs CS+Light. n=6 mice.

neurons. To verify the lesion, we counted the number of neurons (NeuroTrace) in the STN of lesion and control mice and found that the AAV injection reduced the number of STN neurons (*Figure 11A*, Mann-Whitney Z=6.45 $p<0.0001$ Lesion vs Control). The lesion had a strong negative effect on signaled active avoidance tasks (*Figure 11B*) compared to control mice (n=6) and to a group of zona incerta lesion mice from a previous study (*Hormigo et al., 2023*) that underwent the same procedures as the STN lesion mice, but the AAV was injected in Vgat-Cre mice, thereby lesioning GABAergic neurons in the adjacent zona incerta without causing significant impairment in the same tasks. Combining the AA1, AA2, and AA3-CS1 sessions together, we found that the percentage of active avoids was impaired in STN lesion mice compared to both control (ANOVA Tukey t(18) = 5.68, $p=0.002$) and zona incerta lesion mice (Tukey t(18) = 5.04, $p=0.005$). In contrast, the number of ITCs did not differ between the groups (ANOVA F(2,18)=1.04, $p=0.37$), and there was no difference in the percentage of errors in AA3-CS2 between the three groups (ANOVA F(2,18) = 0.23, $p=0.79$). Thus, signaled active avoidance was selectively impaired, as spontaneous crossings and passive avoidance were spared.

We also compared movement during task performance for the three procedures (AA1, AA2, and AA3-CS1) and found that in STN lesion mice compared to control mice, there was an increase in the peak amplitude of the orienting response (ANOVA Tukey t(10)=4.2, $p=0.01$) and a decrease in avoidance interval movement measured either from CS onset (Tukey t(10)=8.3, $p=0.0001$) or from response occurrence (Tukey t(10)=8.3, $p=0.0001$). Finally, the movement during passive avoids in AA3-CS2 trials did not differ between control and lesion mice (*Figure 11D*; Tukey t(10) = 1.5, $p=0.28$). In conclusion, selective lesions of STN glutamatergic lesions impaired signaled active avoidance learning and performance.

Since the AAV-based lesion may leave some STN cells intact, we performed electrolytic lesions of STN (*Figure 11F*), which assures elimination of STN cells. We tested the effect of the lesion in trained mice (n=13; *Figure 11G*) using a repeated measures design, and the lesion mice were also compared to control mice (n=6). The lesion impaired the percentage of active avoids in AA1 (Tukey t(36) = 5.18, $p=0.0043$ before vs after the lesion) and especially in AA2 (Tukey t(36) = 11.7, $p<0.0001$) when ITCs are punished. Mice continued to perform ITCs after the lesion in AA1, albeit at a lower rate (Tukey t(36) = 6.47, $p=0.0003$ AA1 before vs after lesion), and ITCs were further suppressed during AA2 (Tukey t(36) = 7.4, $p<0.0001$ AA1 vs AA2 after lesion). During AA3, which requires discriminating between CS1 and CS2 to select the appropriate action, mice continued to be impaired in active avoidance (AA3-CS1), but the percentage of errors during CS2 in lesion mice was not higher than in control mice (Mixed Anova Tukey t(36)=0.23, $p=0.99$ Control vs Lesion for AA3-CS2).

These results indicate that STN lesions impair signaled active avoidance performance, but do not increase signaled passive avoidance errors. In the context of typical Go/NoGo tasks, STN integrity is important for the Go portion of the task (AA1-3), while the NoGo portion requiring action inhibition or postponement (unsignaled and signaled passive avoidance in AA2/3), and the discrimination (AA3) between the two stimuli signaling the different contingencies does not seem to be dependent on STN.

## STN projections to the midbrain are required for cued active avoidance

The results indicate that glutamatergic STN neurons have a critical role in signaled active avoidance. STN neurons have descending projections to SNr and the tegmentum in the midbrain (*Kita and Kitai, 1987*; *Friedman and Yin, 2023*; *Prasad and Wallén-Mackenzie, 2024*). Thus, we tested the effect of inhibiting the fibers of STN neurons in the midbrain.

To test if the activation of this pathway is necessary for signaled active avoidance, we expressed eArch3.0 in STN-Arch mice (n=11) but placed the optical fibers in the midbrain to inhibit STN fibers coursing to the midbrain in the SNr and midbrain reticular nucleus (mRt). The locations of the optical fiber endings for these groups are depicted in *Figure 12A*. The results were combined because they

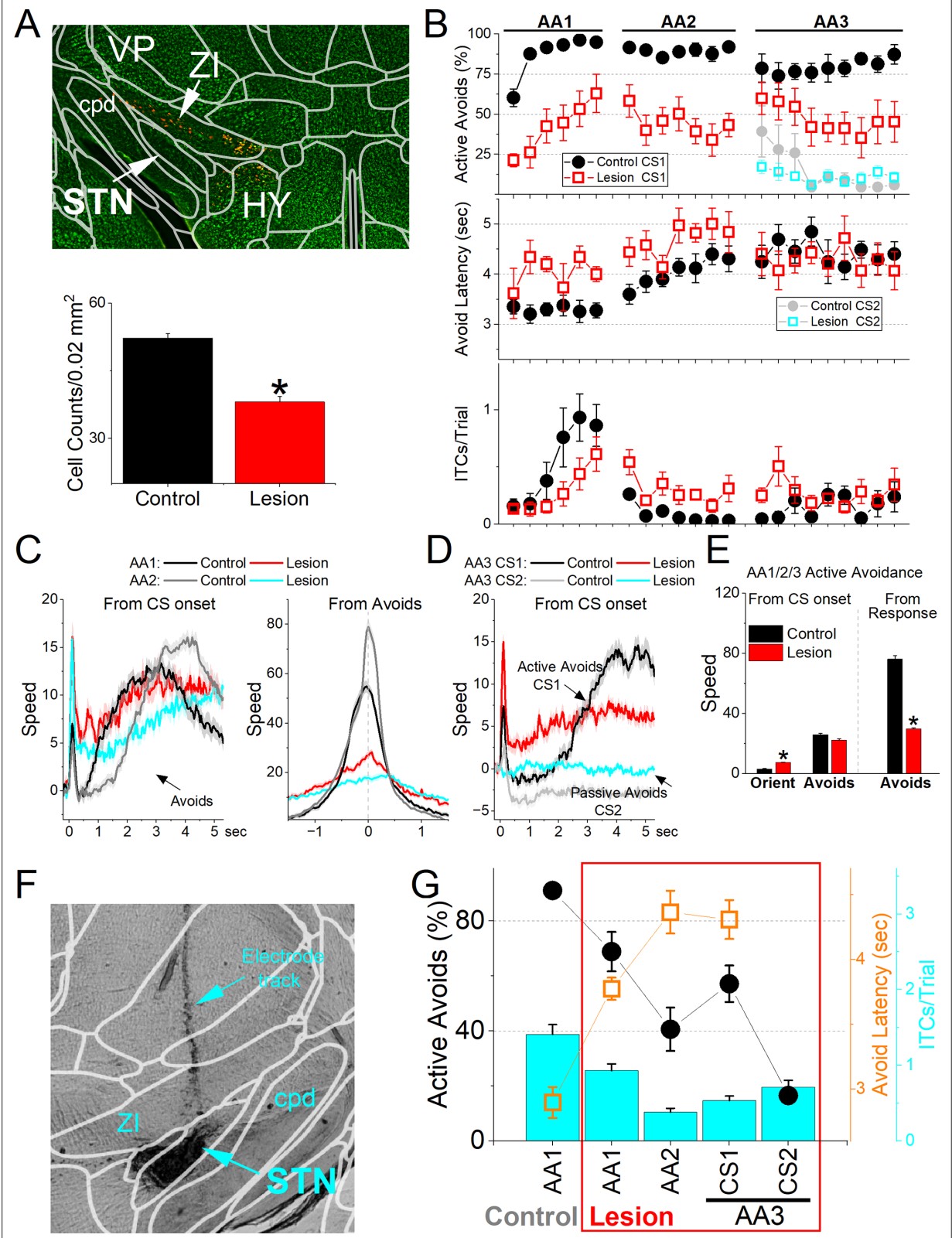

**Figure 11.** Lesions of subthalamic nucleus (STN) glutamatergic neurons impair signaled avoidance learning and performance. (**A**) Coronal Neurotrace (green) stained section of a Vglut2-Cre mouse injected with a Cre-dependent AAV-dtA in the STN to kill glutamatergic neurons. We counted the number of cells in the STN in controls and lesion mice. There was a significant reduction ($p<0.05$) in the number of STN neurons in the lesion mice. (**B**) Behavioral performance during learning of AA1, followed by AA2 and AA3 procedures showing the percentage of active avoids (upper), avoid latency

*Figure 11 continued on next page*

*Figure 11 continued*

(middle), and intertrial crossings (ITCs) (lower) for control and lesion mice. The AA3 procedure shows CS1 and CS2 trials for the same sessions. Active avoids during AA3-CS2 trials are errors, as the mice must passively avoid during CS2. Lesion mice were significantly impaired compared to control mice. (**C**) Movement (speed) from CS onset (left) and from avoid occurrence (right) during AA1 and AA2 procedures for control and lesion mice. The lesion had significant effects on movement measured from CS onset or avoid occurrence. (**D**) Same as *C* for AA3. (**E**) Population measures of orienting, avoidance, and escape responses from CS onset (left) and from response occurrence (right) for overall movement. Asterisks denote significant differences (*p*<0.05) between Control and Lesion. n=12 mice. (**F**) Bilateral electrolytic lesions targeting the STN. (**G**) Effect of bilateral electrolytic STN lesions on behavioral performance in a repeated measures design. The plot shows the percentage of active avoids (filled black circles), avoid latency (open orange squares), and ITCs (cyan bars). Mice were trained in AA1 prior to the lesion and then placed back in AA1, followed by AA2 and AA3. The lesion decreased the percentage of active avoids compared to AA1. During AA2, mice learned to suppress their ITCs. During AA3, lesion mice were impaired in active avoids during CS1 but passively avoided during CS2. n=13 mice.

did not diverge. A group of No Opsin mice (n=5) also had optical fibers in the midbrain. Inhibiting STN fibers in the midbrain with green light during AA1, AA2 and AA3-CS1 CS+Light trials suppressed the percentage of active avoids compared to CS trials in STN-Arch but not in No Opsin mice (*Figure 12B*; Mixed Anova Light × Group; Tukey t(14)=7.03, *p*=0.0008 STN-Arch vs No Opsin for CS+Light trials). Inhibition of STN fibers in the midbrain did not affect escape responses; the mice escaped rapidly at US onset on every failed avoid trial. Furthermore, STN fiber inhibition in the midbrain did not affect the percentage of passive avoids in AA3-CS2 trials (*Figure 12B* right panel; Tukey t(30)=2.5, *p*=0.64 CS vs CS+Light in STN-Arch mice).

Measurements of peak speed (baseline corrected) during AA1, AA2, and AA3-CS1 showed that STN inhibition did not affect the orienting response (*Figure 12C and D*) but there was a suppression of action speed (*Figure 12C and D*; Tukey t(14) = 5.08, *p*=0.01 CS+Light vs CS trials in STN-Arch mice) in association with the abolishment of avoids. However, mice escaped rapidly upon US onset. The peak speed of escapes was not affected by STN fiber inhibition (Tukey t(13) = 0.22, *p*=0.99 CS+Light vs CS trials in STN-Arch mice), indicating that STN inhibition did not paralyze the mice or impair their ability to cross. The movement during the passive avoidance interval in AA3-CS2 trials was somewhat inhibited by STN inhibition (Tukey t(10) = 4.48, *p*=0.04), which would facilitate passive avoidance. These results indicate that STN glutamatergic neuron pathways to the midbrain are essential for signaled active avoidance.

## Discussion

This study examined how STN contributes to goal-directed behavior and movement dynamics by recording and manipulating glutamatergic neuron activity. We found that the STN encodes both movement direction and cue-evoked active avoidance actions, and that its activity is essential for generating these behaviors. Populations of STN neurons were strongly activated during spontaneous contraversive movements and cued active avoidance. After accounting for the movement-related effects using mixed-effects models, STN activity was found to encode cautious avoidance actions characterized by delayed initiation but rapid execution. Notably, a subset of neurons discharged in anticipation of these actions, suggesting a role in planning cautious responses. Other neurons preferentially encoded avoidance errors, responding to aversive, punitive outcomes, suggesting a role in nociceptive processing. Consistent with these functions, irreversible lesions and reversible inactivation of the STN or its midbrain projections abolished signaled active avoidance. Together, these findings identify the STN as a critical hub for executing cued actions by integrating motor, aversive, and cognitive signals to guide cautious, adaptive responding under threat.

### STN encodes directional movement and responds to sensory inputs

We found that STN neurons discharge during self-paced movement onset and active, but not passive, avoidance actions, which is consistent with recent results showing that STN neurons increase firing during self-paced locomotion in head-fixed mice (*Callahan et al., 2024*) and are important for action initiation (*Watson et al., 2021*). Our recordings showed that STN activation is preferentially associated with contraversive movements, indicating a direction-specific contribution of the STN to motor output, although some neurons exhibited a weak ipsiversive bias. Movement was a strong predictor of neural activity during cued avoidance actions and during the presentation of neutral tones, yet sensory signals also emerged after controlling for motor effects. Tone-evoked STN responses, particularly to

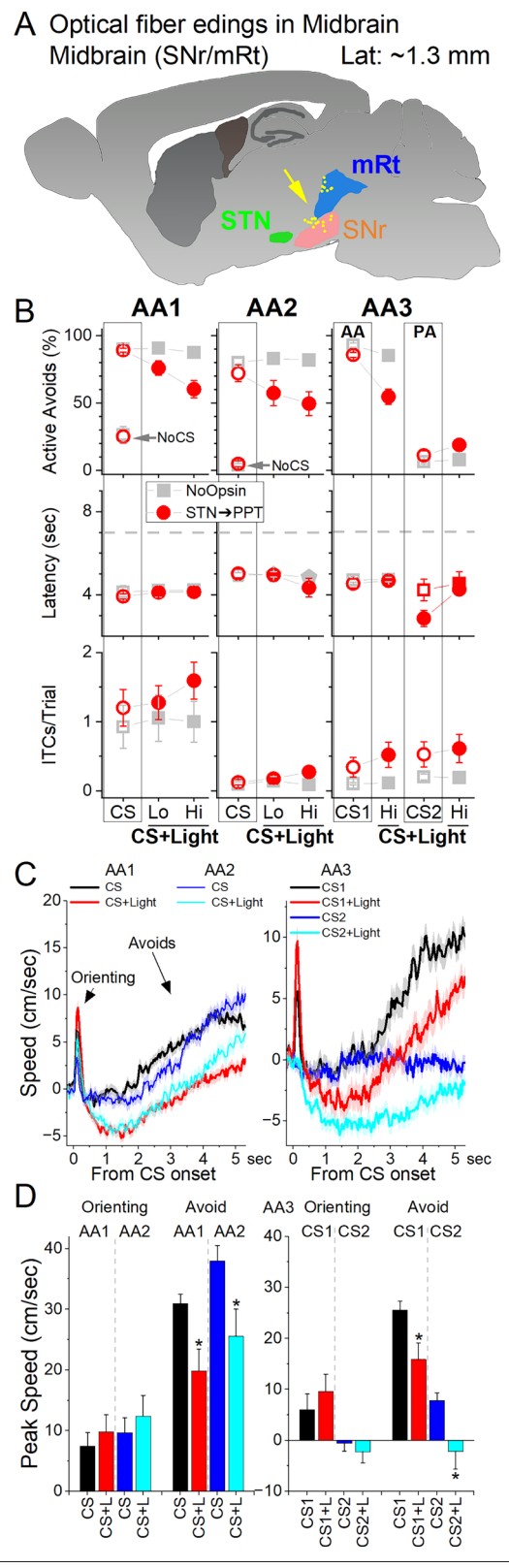

**Figure 12.** Optogenetic inhibition of subthalamic nucleus (STN) glutamatergic fibers in the midbrain impairs signaled avoidance. (**A**) Schematic of optical fiber locations for the midbrain targeting midbrain tegmentum (SNr and mRt) to target fibers originating in STN. (**B**) Effect of Cont green light delivered at different powers (Lo or Hi) on AA1, AA2, and AA3 in mice expressing eArch3.0 in STN glutamatergic neurons. Note the strong abolishment

*Figure 12 continued on next page*

*Figure 12 continued*

of active avoidance responses in CS+Light trials for AA1, AA2, and AA3-CS1. In contrast, passive avoids during AA3-CS2 were not impaired. The light had no effect in No Opsin mice (filled gray squares). (**C**) Traces of overall movement (speed) during AA1, AA2, and AA3 for CS trials and CS+Light trials combined for different light powers. The trials are aligned by CS onset, which reveals the orienting response evoked by the CS followed by the ensuing avoid action. (**D**) Population data of peak speed from CS onset for orienting and avoidance responses during AA1, AA2, and AA3. Asterisks denote significant differences ($p<0.05$) between CS vs CS+Light. n=11 mice.

high-frequency salient auditory stimuli, were dissociable from movement-related activity, indicating that STN neurons integrate both sensory and motor information rather than serving purely motor functions.

## STN encodes aversive outcomes

During actions, population STN activity robustly differentiated behavioral outcomes, showing stronger activation for errors across contingencies. These error-related signals remained significant after controlling for movement and baseline covariates, but their magnitude likely reflects the influence of aversive input. In fact, STN has been linked to pain processing (*Pautrat et al., 2018*; *Jia et al., 2022*; *Serra et al., 2023*). We found stronger activation during unsignaled foot-shock compared to white noise, despite covariate control, indicating a specific contribution of nociceptive processing. This likely accounts for the stronger STN activation observed during active (escapes) and passive avoid errors. Thus, STN activity scaled with escape vigor and painful stimulation, closely linking urgency and nociception, which are difficult to dissociate in behaving animals.

Interestingly, this encoding was primarily confined to groups of neurons that activated weakly (*Type a*, classified by weak activation within active avoidance modes), suggesting that nociceptive processing may occur through distinct neuronal population channels, distinct from those supporting movement and cautious responding.

## STN encodes cautious responding

Intriguingly, the strongest activation of STN neurons, independent of movement and baseline covariates, was observed during the most delayed, cautious responses rather than rapid-onset actions. Caution responding is a hallmark of avoidance behavior, contrasting with appetitive actions, and becomes more pronounced under demanding situations (*Zhou et al., 2022*). In humans, low-frequency oscillatory activity from the prefrontal cortex to the STN increases decision thresholds, promoting more deliberative and less impulsive actions (*Frank et al., 2007*). Conversely, high-frequency STN stimulation in Parkinson's patients disrupts these signals, lowers decision thresholds, and induces impulsive decisions (*Cavanagh et al., 2011*; *Herz et al., 2018*; *Herz et al., 2024*; *Pagnier et al., 2024*).

We found that pre-cue STN activity biased upcoming performance differently depending on the neuronal population. Elevated baseline firing in neurons insensitive to movement predicted avoidance errors, whereas higher pre-cue discharge in movement-correlated neurons predicted correct avoidance actions. During the action phase, population STN activation was stronger for the most cautious actions, independent of movement and baseline covariates. This activation was driven by a subgroup of neurons that either closely followed or anticipated the action. The anticipating neurons may contribute to the delayed onset of cautious responding by transiently suppressing or slowing action initiation.

Together, these results demonstrate that STN coding of cautious behavior is not uniform across the population but instead emerges from the coordinated activity of distinct neuronal subtypes. In addition to movement and nociceptive channels, the STN contains a neuronal processing channel encoding cognitive signals related to caution and decision control, functioning as a heterogeneous hub where sensory, motor, and cognitive information converge to guide adaptive avoidance under uncertainty and threat.

## STN is essential for cued avoidance actions

A key finding of our study is that optogenetic inhibition of STN neurons impairs signaled active avoidance actions without significantly affecting the ability of mice to escape the US, despite both

behaviors requiring the same shuttling movement. This effect was corroborated by results from both AAV-mediated and electrolytic STN lesions. Furthermore, inhibition of STN projections to the midbrain tegmentum also impaired active avoidance. These findings demonstrate that STN activation is critical for generating cued goal-directed actions, supporting the idea that the STN functions as an action center working in coordination with the midbrain tegmentum, which is likewise essential for mediating these goal-directed behaviors (*Hormigo et al., 2019*).

The finding that STN is essential for cued active avoidance actions is underscored by the observation that similar methods to inhibit or lesion related brain areas, such as zona incerta (*Hormigo et al., 2020*; *Hormigo et al., 2023*), nucleus accumbens (*Zhou et al., 2024*), or STN's target in SNr (*Hormigo et al., 2016*; *Hormigo et al., 2021b*; *Hormigo et al., 2021c*), do not impair these behaviors. While these regions can modulate avoidance responses via their direct GABAergic inhibitory projections to the midbrain tegmentum, they are not essential for mediating these actions. Moreover, since SNr inhibition does not suppress signaled avoidance (*Hormigo et al., 2021c*), it is unlikely that the STN generates avoidance responses by robustly exciting the SNr, as this strongly suppresses active avoidance. Instead, our findings support a model in which the STN drives actions through its direct projections to the midbrain tegmentum. This does not preclude the STN from generating other movements, such as self-paced actions, through its connections with the SNr (*Klaus et al., 2019*). However, the execution of cued avoidance actions—characterized by slow onset timing—appears to specifically depend on STN's direct pathways to the midbrain tegmentum.

A potential concern is whether increased avoidance latency under some conditions should be interpreted as heightened caution. This interpretation is valid only when animals are performing active avoidance behavior normally. In intact animals, avoidance success remains high across tasks (AA1-3), and the CS1 trials used to measure latency are identical across tasks. In this stable behavioral context, shifts in latency reflect adaptive adjustments in caution, because actions remain tightly linked to the cue and the trial rules are unchanged. However, this logic does not apply when the STN function is disrupted. STN inhibition or lesions collapse avoidance performance to near-chance levels, and the few crossings that occur are often poorly aligned to the cue and likely reflect random movement rather than an intentional avoidance response. Once behavior breaks down, latency no longer reflects the same processes. A helpful analogy is a pedestrian crossing the street: waiting longer before stepping off the curb in heavy traffic reflects caution but moving slowly because of impairment while failing to cross on many trials does not. Likewise, latency can be interpreted as caution only when avoidance behavior is intact, goal-directed, and testing conditions remain unchanged. Under STN inactivation, where cued avoidance fails, longer latencies cannot be taken as evidence of caution.

While viral and electrolytic lesions resulted in deficits in cued active avoidance, the ability to withhold responses during the intertrial interval, when uncued actions are punished in AA2, remained mostly unaltered. Likewise, these lesions did not increase error rates during CS2 in AA3. This preservation of passive actions might reflect a default NoGo state when STN is inhibited, consistent with its role in cued active avoidance actions. It is noteworthy that we did not investigate conditions where previously triggered actions must be cancelled (*Aron, 2011*), partly due to our finding that inhibiting STN suppresses the initiation of cued actions. Cued avoidance actions can be set up to test this directly in future work. Overall, our results support the role of the STN in initiating and regulating the timing of movements in response to learned cues. The preserved passive avoidance suggests that, while the STN is crucial for initiating actions in response to potential threats, it may not be essential for behaviors that rely on default response withholding.

Our results using viral and electrolytic lesions, as well as optogenetic inhibition of STN neurons, show that signaled active avoidance is virtually abolished, and this deficit is reproduced when we selectively inhibit STN fibers in the midbrain. Inhibition of STN projections in either the SNr or mRt eliminates cued avoidance responses. Critically, mice continue to escape during US presentations after lesions or during photoinhibition, indicating that basic motor capabilities and rapid defensive actions remain preserved. These findings argue against the idea that the STN's role in avoidance arises from a nonspecific suppression or facilitation of motor tone, even if the STN also contributes to general movement control. Instead, the loss-of-function data demonstrate that STN output is required specifically for generating cued actions that depend on interpreting sensory information and applying learned contingencies to decide when to act. Thus, while STN activity can modulate movement

parameters, its essential role in cued avoidance may reflect a selective contribution to goal-directed action selection under threat rather than a generic motor modulation.

We propose that STN neuron projections to the midbrain tegmentum are essential for mediating cued avoidance actions, functioning independently of other projections. While the activation imposed by the STN on the SNr during cued actions may serve a modulatory but non-essential role, its direct projections to the midbrain tegmentum may be the critical pathway for generating these actions.

## Implications for STN function in adaptive behavior

Our findings demonstrate that STN circuits are essential for cued active avoidance actions, establishing the STN as a central forebrain hub in adaptive avoidance circuitry that integrates sensory salience, action vigor, aversive signals, and cognitive information to shape defensive behavior. The data support a model in which the STN contributes to cued avoidance by coordinating timed actions in response to threat-predictive cues through its projections to the midbrain. Its role extends beyond simple motor gating, as STN activity integrates nociceptive and movement-related inputs with cognitive signals related to caution, allowing the system to balance urgency and deliberation under threat. Unlike other subthalamus or basal ganglia nuclei that may represent avoidance actions and encode similar signals, the STN is necessary for generating them. These findings deepen our understanding of how brain circuits generate adaptive motor responses to avoid harm, a fundamental behavior conserved across species, including humans.

# Materials and methods
## Experimental design and statistical analysis

The methods used in the present paper were like those employed in our previous studies (e.g. *Hormigo et al., 2023*; *Zhou et al., 2024*). This study was performed in strict accordance with the recommendations in the Guide for the Care and Use of Laboratory Animals of the National Institutes of Health. All of the animals were handled according to approved institutional animal care and use committee (IACUC) protocols of UConn Health (Animal Welfare Assurance Number: A3471-01). The protocol was approved by the Committee on the Ethics of Animal Experiments of UConn Health. All surgery was performed under isoflurane anesthesia, and every effort was made to minimize suffering. All procedures were conducted in adult (>8 weeks) male and female mice. Most experiments used a repeated-measures design in which each mouse or cell served as its own control (within-subject/cell comparisons), but we also compared experimental groups between subjects or cells (between-group comparisons). To test the main effects of experimental variables, we used either repeated-measures ANOVA or linear mixed-effects models.

Mixed-effects models were used to quantify relationships between STN $\Delta F/F$ activity (dependent variable) and experimental factors while accounting for repeated measures. Models included fixed effects for task contingency, outcome, cell/avoid classes, etc., and their interactions, along with window-specific covariates for head speed and baseline $\Delta F/F$ (centered using z-scores as noted in the results). Random effects were specified as sessions nested within subjects or cells nested within subjects, following the general syntax: $\Delta F/F \sim$ (Factor1 * Factor2 * …) * Speed + (1 | Subject/Session) in lme4. Separate models were fit for each of the four behavioral windows, using area measures for the baseline, orienting, avoidance, and from-action windows (normalized by window duration), with the latter windows corrected by the baseline. Generally, the baseline window ($-0.5$–$0$ s from CS onset) captures the pre-cue state at trial initiation, whereas the orienting window ($0$–$0.5$ s post-CS) reflects the initial response to the cue. The avoidance window (with task-specific durations) corresponds to the period when mice must generate or withhold a response to avoid punishment, excluding both the preceding orienting window and the subsequent escape interval. The from-action window aligns time series from the occurrence of a response (e.g. avoid, escape, passive avoid error), enabling comparison between avoidance responses occurring within the avoidance interval and escape responses in the later interval. Including the window-specific head speed covariates, along with baseline $\Delta F/F$ and head speed measures, in the post-CS window models allowed us to dissociate movement-related modulation from neural encoding of task-related factors. Model fit was assessed by likelihood ratio tests comparing nested models, confirming that inclusion of covariates and interactions improved explanatory power. For significant main effects or interactions, post-hoc pairwise comparisons were

performed using *emmeans* with Tukey's correction (Holm-adjusted). For continuous covariates, slope effects were evaluated using *emtrends*, with significance determined by t-tests ($p<0.05$).

To enable rigorous approaches, we maintain a centralized metadata system that logs all details about the experiments and is engaged for data analyses (*Castro-Alamancos, 2022*). Moreover, during daily behavioral sessions, computers run experiments automatically using preset parameters logged for reference during analysis. Analyses are performed using scripts that automate all aspects of data analysis from access to logged metadata and data files to population statistics and graph generation.

## Strains and adeno-associated viruses (AAVs)

To record from glutamatergic STN neurons using calcium imaging, we injected a Cre-dependent AAV AAV5-syn-FLEX-jGCaMP7f-WPRE (Addgene: $7\times10^{12}$ vg/ml) in the STN of Vglut2-cre mice (Jax 028863; B6J.129S6(FVB)-Slc17a6$^{tm2(cre)Lowl}$/MwarJ) to express GCaMP6f/7 f. An optical fiber or GRIN lens was then placed in this location. To inhibit glutamatergic STN neurons using optogenetics, we expressed eArch3.0 by injecting AAV5-EF1a-DIO-eArch3.0-EYFP (UNC Vector Core, titers: $3.4\times10^{12}$ vg/ml) in the STN of Vglut2-cre mice (STN-Arch mice). To kill glutamatergic STN neurons, we injected AAV8-EF1a-mCherry-flex-dtA (Neurophotonics: $1.3\times10^{13}$ GC/ml) into the STN of Vglut2-cre mice. No-Opsin controls were injected with AAV8-hSyn-EGFP (Addgene, titers: $4.3\times10^{12}$ GC/ml by quantitative PCR) or nil in the STN. For optogenetics, we implanted dual optical fibers bilaterally in the STN or its projection targets. All the optogenetic methods used in the present study have been validated in previous studies using slice and/or in vivo electrophysiology (*Hormigo et al., 2016*; *Hormigo et al., 2019*; *Hormigo et al., 2021b*; *Hormigo et al., 2021c*).

## Surgeries

Optogenetics and fiber photometry experiments involved injecting 0.2–0.4 µl AAVs per site during isoflurane anesthesia (~1%). Animals received carprofen after surgery. The stereotaxic coordinates for injection in STN are (from bregma; lateral from the midline; ventral from the bregma-lambda plane in mm): 2.1 posterior; 1.7; 4.2. In these experiments, a single (400 µm in diameter for fiber photometry or 600 µm lens for miniscope) or dual (200 µm in diameter for optogenetics) optical fiber was implanted unilaterally or bilaterally during isoflurane anesthesia. The stereotaxic coordinates for the implanted optical fibers (in mm) are: STN (2–2.1 posterior; 1.5; 4.2–4.3), and midbrain (3.3–3.7 posterior; 1.5; 2.9–4.1). The coordinate ranges reflect different animals that were combined because the coordinate differences produced similar effects. No Opsin mice were implanted with cannulas in STN or its projection sites, and the results were combined after confirming that light produced similar effects in these animals.

## Active avoidance tasks

Mice were trained in a signaled active avoidance task, as previously described (*Hormigo et al., 2016*; *Hormigo et al., 2019*). During an active avoidance session, mice are placed in a standard shuttle box (16.1"×6.5") that has two compartments separated by a partition with side walls forming a doorway that the animal must traverse to shuttle between compartments. A speaker is placed on one side, but the sound fills the whole box and there is no difference in behavioral performance (signal detection and response) between sides. A trial consists of a 7 s avoidance interval followed by a 10 s escape interval. During the avoidance interval, an auditory CS (8 kHz 85 dB) is presented for the duration of the interval or until the animal produces a conditioned response (avoidance response) by moving to the adjacent compartment, whichever occurs first. If the animal avoids, by moving to the next compartment, the CS ends, the escape interval is not presented, and the trial terminates. However, if the animal does not avoid, the escape interval ensues by presenting white noise and a mild scrambled electric foot-shock (0.3 mA) delivered through the grid floor of the occupied half of the shuttle box. This unconditioned stimulus (US) readily drives the animal to move to the adjacent compartment (escape response), at which point the US terminates, and the escape interval and the trial end. Thus, an *avoidance response* will eliminate the imminent presentation of a harmful stimulus. An *escape response* is driven by the presentation of the harmful stimulus to eliminate the harm it causes. Successful avoidance warrants the absence of harm. Each trial is followed by an intertrial interval (duration is randomly distributed; 25–45 s range), during which the animal awaits the next trial. Mice performed 50–100 trials per daily session. The initial session is a habituation session lasting

20 min to explore the cage. We employed four variations of the basic signaled active avoidance procedure termed AA1, AA2, AA3, and AA4 presented sequentially in daily sessions (seven sessions per animal).

In AA1, mice are free to cross between compartments during the intertrial interval; there is no consequence for intertrial crossings (ITCs).

In AA2, mice receive a 0.2 s foot-shock (0.3 mA) and white noise for each ITC. Therefore, in AA2, mice must passively avoid during the intertrial interval by inhibiting their tendency to shuttle between trials, termed intertrial crossings (ITCs). Thus, during AA2, mice perform both signaled active avoidance during the signaled avoidance interval (like in AA1) and unsignaled passive avoidance during the unsignaled intertrial interval.

In AA3, mice are subjected to a CS discrimination procedure in which they must respond differently to a CS1 (8 kHz tone at 85 dB) and a CS2 (4 kHz tone at 75 dB) presented randomly (half of the trials are CS1). Mice perform the basic signaled active avoidance to CS1 (like in AA1 and AA2), but also perform signaled passive avoidance to CS2, and ITCs are not punished. In AA3, if mice shuttle during the CS2 avoidance interval (7 s), they receive a 0.5 s foot-shock (0.3 mA) with white noise and the trial ends. If animals do not shuttle during the CS2 avoidance interval, the CS2 trial terminates at the end of the avoidance interval (i.e. successful signaled passive avoidance).

In AA4, three different CSs, CS1 (8 kHz tone at 85 dB), CS2 (10 kHz tone at 85 dB), and CS3 (12 kHz tone at 85 dB) signal a different avoidance interval duration of 4, 7, and 15 s, respectively. Like in AA2, mice are punished for producing intertrial crossings. In AA4, mice adjust their response latencies according to the duration of the avoidance interval signaled by each CS.

In a modified version of AA1, AA2, and AA3, we introduced randomized presentations of CS1 (8 kHz), CS2 (4 kHz), and CS3 (12 kHz) at three different saliency levels (65, 75, and 85 dB), starting from the first AA1 session. This was employed in the miniscope experiments to make the tasks more difficult, increasing the number of errors. In this design, CS1 predicted the aversive US and required an active avoidance response, while in AA1/2, CS2 and CS3 were neutral—they predicted nothing. However, crossings during any CS turned off the tone, and crossings during CS1 also avoided the US. In AA2, ITCs were punished, and in AA3, mice were required to passively avoid (not cross) during CS2 to avoid the US while CS3 continued to be neutral.

There are three main variables representing task performance. The percentage of active avoidance responses (% avoids) represents the trials in which the animal actively avoided the US in response to the CS. The response latency (latency) represents the time (s) at which the animal enters the safe compartment after the CS onset; avoidance latency is the response latency only for successful active avoidance trials (excluding escape trials). The number of crossings during the ITCs represents random shuttling due to locomotor activity in the AA1 and AA3 procedures, or failures to passively avoid in the AA2 procedure. The sound pressure level (SPL) of the auditory CSs was measured using a microphone (PCB Piezotronics 377C01) and amplifier (100x) connected to a custom LabVIEW application that samples the stimulus within the shuttle cage as the microphone rotates driven by an actuator controlled by the application.

## Fiber photometry

We employed a 2-channel excitation (465 and 405 nm) and 2-channel emission (525 and 430 nm for GCaMP6f and other emissions) fiber photometry system (Doric Lenses). Alternating light pulses were delivered at 100 Hz (per each 10 ms, 465 is on for 3 ms, and 2 ms later, 405 is on for 3 ms). While monitoring the 525 nm emission channel, we set the 465 light pulses in the 20–60 µW power range, and then the power of the 405 light pulses was adjusted (20–50 µW) to approximately match the response evoked by the 465 pulses. During recordings, the emission peak signals evoked by the 465 (GCaMP6f) and 405 (isobestic) light pulses were acquired at 5–20 kHz and measured at the end of each pulse. To calculate $F_o$, the measured peak emissions evoked by the 405 nm pulses were scaled to those evoked by the 465 pulses (F) using the slope of the linear fit. Finally, ΔF/F was calculated with the following formula: $(F-F_o)/F_o$ and converted to Z-scores. Due to the nature of the behavior studied, a swivel is essential. We employed a rotatory-assisted photometry system that has no light path interruptions (Doric Lenses). In addition, charcoal powder was mixed in the dental cement to assure that ambient light was not leaking into the implant and reaching the optical fiber; this was tested in each animal by comparing fluorescence signals in the dark versus normal cage illumination.

## Miniscope imaging

We employed GRIN lenses (0.6 mm diameter, 7.3 mm length) and an nVista (Inscopix) recording system coupled with an electrical swivel. We added custom tungsten rods to the GRIN lenses to maximize the stability of the recordings. During each session, we adjusted the focus plane to record the best neurons, which were assigned as independent neurons per session. We used Inscopix Data Processing Software (IDPS) to extract ROIs from the raw miniscope movies. Briefly, the movies were preprocessed with a spatial bandpass filter, which removes the low and high spatial frequency content from the movies, minimizing out-of-plane neuropil fluorescence and allowing visual identification of putative neurons. The movies were then motion corrected, using the initial frame as the global reference. Finally, neurons were manually outlined on the processed movie, and the ΔF/F calcium activity was calculated from these ROIs. To assure stability during each recording session, the ROIs were visually verified from start to end for each video.

K-means clustering (scikit-learn) was performed on features extracted from ΔF/F (z-score), movement (speed), or cross-correlation time series traces. These features included peak amplitudes, areas under the curve, and times to peak around specific events (e.g. turns, CS onset, or avoidance occurrence). Alternatively, clustering was also applied to principal component scores derived from the same traces, which were treated as an analog of spectral data for PCA.

## Optogenetics

The implanted optical fibers were connected to patch cables using sleeves. A black aluminum cap covered the head implant and completely blocked any light exiting at the ferrule's junction. Furthermore, the experiments occurred in a brightly lit cage that made it difficult to detect any light escaping the implant. The other end of the patch cable was connected to a dual light swivel (Doric lenses) that was coupled to a green laser (520 nm; 100 mW) to activate Arch. In experiments expressing Arch, green light was applied continuously at different powers (3.5, 7, 15, 25, and 30 mW). Power is regularly measured by flashing the connecting patch cords onto a light sensor—with the sleeve on the ferrule.

During optogenetic experiments that involve avoidance procedures, we compared different trial types: CS and CS+Light. *CS trials* were standard avoidance trials specific to each procedure, without optogenetic stimulation. *CS+Light trials* were identical to CS trials, except that optogenetic light was delivered simultaneously with the CS and the US during the avoid and escape intervals. To perform within-group repeated measures (RM) comparisons, the different trial types for a procedure were delivered randomly within the same session. In addition, the trials were compared between different groups, including No Opsin mice that did not express opsins but were subjected to the same trials, including light delivery.

## Video tracking

All mice in the study (open field or shuttle box) were continuously video tracked (30–100 FPS) in synchrony with the procedures and other measures. During open field experiments, mice are placed in a circular open field (10" diameter) that was illuminated from the bottom or in the standard shuttle box (16.1"×6.5"). We automatically tracked head movements with two color markers attached to the head connector – one located over the nose and the other between the ears. The coordinates from these markers form a line (head midline) that serves to derive several instantaneous movement measures per frame (*Zhou et al., 2023*). Overall head movement was separated into *rotational* and *translational* components (unless otherwise indicated, overall head movement is presented for simplicity and brevity, but the different components were analyzed). Rotational movement was the angle formed by the head midline between succeeding video frames multiplied by the radius. Translational movement resulted from the sum of linear (forward vs backward) and sideways movements. *Linear* movement was the distance moved by the ears marker between succeeding frames multiplied by the cosine of the angle formed by the line between these succeeding ear points and the head midline. *Sideways* movement was calculated as linear movement, but the sine was used instead of the cosine. Pixel measures were converted to metric units using calibration and expressed as speed (cm/s). We used the time series to extract window measurements around events (e.g. CS presentations). Measurements were obtained from single-trial traces and/or from traces averaged over a session. In addition, we obtained the direction of the rotational movement with a *Head Angle* or *bias* measure, which was

the accumulated change in angle of the head per frame (versus the previous frame) zeroed by the frame preceding the stimulus onset or event (this is equivalent to the rotational speed movement in degrees). The *time to peak* is when the *extrema* occurs versus event onset.

To detect spontaneous turns or movements from the head tracking, we applied a local maximum algorithm to the continuous head angle or speed measure, respectively. Every point is checked to determine if it is the maximum or minimum among the points in a range of 0.5 s before and after the point. A change in angle of this point >10 degrees was a detected turn in the direction of the sign. We further sorted detected turns or movements based on the timing of previous detected events.

## Histology

Mice were deeply anesthetized with an overdose of isoflurane. Upon losing all responsiveness to a strong tail pinch, the animals were decapitated, and the brains were rapidly extracted and placed in fixative. The brains were sectioned (100 μm sections) in the coronal or sagittal planes. Some sections were stained using NeuroTrace. All sections were mounted on slides, cover-slipped with DAPI mounting media, and all the sections were imaged using a slide scanner (Leica Thunder). We used an APP we developed with OriginLab (Brain Atlas Analyzer) to align the sections with the Allen Brain Atlas Common Coordinate Framework (CCF) v3 (*Wang et al., 2020*). This reveals the location of probes and fluorophores versus the delimited atlas areas. We used it to determine the extent of the lesions by delimiting DAPI or NeuroTrace-stained sections of control, dtA, and electrolytic lesion mice. We counted NeuroTrace-stained neurons within the STN using image stacks acquired with Leica Thunder using LAS X software.

## Acknowledgements

Supported by NIH grants to MAC. We thank Mariana Mangini for technical assistance.

## Additional information

### Funding

| Funder | Grant reference number | Author |
| --- | --- | --- |
| National Institute of Neurological Disorders and Stroke | R35NS097272 | Manuel A Castro-Alamancos |
| National Institute of Neurological Disorders and Stroke | R01NS104810 | Manuel A Castro-Alamancos |

The funders had no role in study design, data collection and interpretation, or the decision to submit the work for publication.

### Author contributions

Ji Zhou, Muhammad Sarmad Sajid, Sebastian Hormigo, Data curation, Formal analysis, Investigation, Writing – review and editing; Manuel A Castro-Alamancos, Conceptualization, Data curation, Formal analysis, Supervision, Funding acquisition, Investigation, Methodology, Writing – original draft, Project administration, Writing – review and editing

### Author ORCIDs

Sebastian Hormigo ⓘ https://orcid.org/0000-0003-3141-5965
Manuel A Castro-Alamancos ⓘ https://orcid.org/0000-0002-2916-9585

### Ethics

This study was performed in strict accordance with the recommendations in the Guide for the Care and Use of Laboratory Animals of the National Institutes of Health. All of the animals were handled according to approved institutional animal care and use committee (IACUC) protocols of UConn Health (Animal Welfare Assurance Number: A3471-01). The protocol was approved by the Committee

on the Ethics of Animal Experiments of UConn Health. All surgery was performed under isoflurane anesthesia, and every effort was made to minimize suffering.

Reviewer #1 (Public review): https://doi.org/10.7554/eLife.107796.4.sa1
Reviewer #2 (Public review): https://doi.org/10.7554/eLife.107796.4.sa2
Reviewer #3 (Public review): https://doi.org/10.7554/eLife.107796.4.sa3
Author response https://doi.org/10.7554/eLife.107796.4.sa4

---

# Additional files

## Supplementary files
MDAR checklist

## Data availability
All data have been deposited in Dryad (https://doi.org/10.5061/dryad.0rxwdbsfp).

The following dataset was generated:

| Author(s) | Year | Dataset title | Dataset URL | Database and Identifier |
|---|---|---|---|---|
| Castro-Alamancos MA, Zhou J, Sajid M, Hormigo S, Castro-Alamancos MA | 2026 | A Forebrain Hub for Cautious Actions via the Midbrain | https://doi.org/10.5061/dryad.0rxwdbsfp | Dryad Digital Repository, 10.5061/dryad.0rxwdbsfp |

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
