## [Editor Report · eLife Assessment]

This **valuable** study uses fiber photometry, implantable lenses, and optogenetics, to show that a subset of subthalamic nucleus neurons are active during movement, and that active but not passive avoidance depends in part on STN projections to substantia nigra. The strength of the evidence for these claims is **solid** and this paper may be of interest to basic and applied behavioural neuroscientists working on movement or avoidance.

---

## [Referee Report · Reviewer #1 (Public review)]

Summary:

The manuscript presents a robust set of experiments that provide new insights into the role of STN neurons during active and passive avoidance tasks. These forms of avoidance have received comparatively less attention in the literature than the more extensively studied escape or freezing responses, despite being extremely relevant to human behaviour and more strongly influenced by cognitive control.

Strengths:

Understanding the neural infrastructure supporting avoidance behaviour would be a fundamental milestone in neuroscience. The authors employ sophisticated methods to delineate the role of STN neurons during avoidance behaviours. The work is thorough and the evidence presented is compelling. Experiments are carefully constructed, well-controlled, and the statistical analyses are appropriate.

---

## [Referee Report · Reviewer #2 (Public review)]

Summary:

Zhou, Sajid et al. present a study investigating the STN involvement in signaled movement. They use fiber photometry, implantable lenses, and optogenetics during active avoidance experiments to evaluate this. The data are useful for the scientific community and the overall evidence for their claims is solid, but many aspects of the findings are confusing. The authors present a huge collection of data, it is somewhat difficult to extract the key information and the meaningful implications resulting from these data.

Strengths:

The study is comprehensive in using many techniques and many stimulation powers and frequencies and configurations.

---

## [Referee Report · Reviewer #3 (Public review)]

Summary:

The authors use calcium recordings from STN to measure STN activity during spontaneous movement and in a multi-stage avoidance paradigm. They also use optogenetic inhibition and lesion approaches to test the role of STN during the avoidance paradigm. The paper reports a large amount of data and makes many claims, some seem well supported to this Reviewer, others not so much.

Strengths:

Well-supported claims include data showing that during spontaneous movements, especially contraversive ones, STN calcium activity is increased using bulk photometry measurements. Single-cell measures back this claim but also show that it is only a minority of STN cells that respond strongly, with most showing no response during movement, and a similar number showing smaller inhibitions during movement.

Photometry data during cued active avoidance procedures show that STN calcium activity sharply increases in response to auditory cues, and during cued movements to avoid a footshock. Optogenetic and lesion experiments are consistent with an important role for STN in generating cue-evoked avoidance. And a strength of these results is that multiple approaches were used.

[Editors' note: The authors provided a good explanation regarding the difference between interpreting 'caution' in the healthy vs impaired situation, and this addressed one of the remaining major concerns from the last round of review.]

---

## [Author Response]

The following is the authors’ response to the previous reviews

**Public Reviews:**

**Reviewer #1 (Public review):**
One possible remaining conceptual concern that might require future work is determining whether STN primarily mediates higher-level cognitive avoidance or if its activation primarily modulates motor tone.

Our results using viral and electrolytic lesions (Fig. 11) and optogenetic inhibition of STN neurons (Fig. 10) show that signaled active avoidance is virtually abolished, and this effect is reproduced when we selectively inhibit STN fibers in the midbrain (Fig. 12). Inhibition of STN projections in either the substantia nigra pars reticulata (SNr) or the midbrain reticular tegmentum (mRt) eliminates cued avoidance responses while leaving escape responses intact. Importantly, mice continue to escape during US presentation after lesions or during photoinhibition, demonstrating that basic motor capabilities and the ability to generate rapid defensive actions are preserved.

These findings argue against the idea that STN’s role in avoidance reflects a nonspecific suppression or facilitation of motor tone, even if the STN also contributes to general movement control. Instead, they show that STN output is required for generating “cognitively” guided cued actions that depend on interpreting sensory information and applying learned contingencies to decide when to act. Thus, while STN activity can modulate movement parameters, the loss-of-function results point to a more selective role in supporting cued, goal-directed avoidance behavior rather than a general adjustment of motor tone.

**Reviewer #2 (Public review):**
All previous weaknesses have been addressed. The authors should explain how inhibition of the STN impairing active avoidance is consistent with the STN encoding cautious action. If 'caution' is related to avoid latency, why does STN lesion or inhibition increase avoid latency, and therefore increase caution? Wouldn't the opposite be more consistent with the statement that the STN 'encodes cautious action'?

The reviewer’s interpretation treats any increase in avoidance latency as evidence of “more caution,” but this holds only when animals are performing the avoidance behavior normally. In our intact animals, avoidance rates remain high across AA1 → AA2 → AA3, and the active avoidance trials (CS1) used to measure latency are identical across tasks (e.g., in AA2 the only change is that intertrial crossings are punished). Under these conditions, changes in latency genuinely reflect adjustments in caution, because the behavior itself is intact, actions remain tightly coupled to the cue, and the trials are identical.

This logic does not apply when STN function is disrupted. STN inhibition or lesions reduce avoidance to near chance levels; the few crossings that do occur are poorly aligned to the CS and many likely reflect random movement rather than a cued avoidance response. Once performance collapses, latency can no longer be assumed to reflect the same cognitive process. Thus, interpreting longer latencies during STN inactivation as “more caution” would be erroneous, and we never make that claim.

A simple analogy may help clarify this distinction. Consider a pedestrian deciding when to cross the street after a green light. If the road is deserted (like AA1), the person may step off the curb quickly. If the road is busy with many cars that could cause harm (like AA2), they may wait longer to ensure that all cars have stopped. This extra hesitation reflects *caution*, not an inability to cross. However, if the pedestrian is impaired (e.g., cannot clearly see the light, struggles to coordinate movements, or cannot reliably make decisions), a delayed crossing would not indicate greater caution—it would reflect a breakdown in the ability to perform the behavior itself. The same principle applies to our data: we interpret latency as “caution” only when animals are performing the active avoidance behavior normally, success rates remain high, and the trial rules are identical. Under STN inhibition or lesion, when active avoidance collapses, the latency of the few crossings that still occur can no longer be interpreted as reflecting caution. We have added these points to the Discussion.

**Reviewer #3 (Public review):**
Original Weaknesses:I found the experimental design and presentation convoluted and some of the results over-interpreted.

We appreciate the reviewer’s comment, but the concern as stated is too general for us to address in a concrete way. The revised manuscript has been substantially reorganized, with simplified terminology, streamlined figures, and removal of an entire set of experiments to avoid over-interpretation. We are confident that the experimental design and results are now presented clearly and without extrapolation beyond the data. If there are specific points the reviewer finds convoluted or over-interpreted, we would be happy to address them directly.

As presented, I don't understand this idea that delayed movement is necessarily indicative of cautious movements. Is the distribution of responses multi-modal in a way that might support this idea; or do the authors simply take a normal distribution and assert that the slower responses represent 'caution'? Even if responses are multi-modal and clearly distinguished by 'type', why should readers think this that delayed responses imply cautious responding instead of say: habituation or sensitization to cue/shock, variability in attention, motivation, or stress; or merely uncertainty which seems plausible given what I understand of the task design where the same mice are repeatedly tested in changing conditions. This relates to a major claim (i.e., in the title).

We appreciate the reviewer’s question and address each component directly.

(1) What we mean by “caution” and how it is operationalized

In our study, caution is defined operationally as a systematic increase in avoidance latency when the behavioral demand becomes higher, while the trial structure and required response remain unchanged. Specifically, CS1 trials are identical in AA1, AA2, and AA3. Thus, when mice take longer to initiate the same action under more demanding contexts, the added time reflects additional evaluation before acting—consistent with longestablished interpretations of latency shifts in cognitive psychology (see papers by Donders, Sternberg, Posner) and interpretations of deliberation time in speed-accuracy tradeoff literature.

(2) Why this interpretation does not rely on multi-modal response distributions We do not claim that “cautious” responses form a separate mode in the latency distribution. The distributions are unimodal, and caution is inferred from conditiondependent shifts in these distributions across identical trials, not from the existence of multiple peaks (see Zhou et al, 2022). Latency shifts across conditions with identical trial structure are widely used as behavioral indices of deliberation or caution.

(3) Why alternative explanations (habituation/sensitization, motivation, attention, stress, uncertainty) do not account for these latency changes

Importantly, nothing changes in CS1 trials between AA1 and AA2 with respect to the cue, shock, or required response. Therefore:

- Habituation/sensitization to the cue or shock cannot explain the latency shift (the stimuli and trial type are unchanged). We have previously examined cue-evoked orienting responses and their habituation in detail (Zhou et al., 2023), and those measurements are dissociable from the latency effects described here.

- Motivation or attention are unlikely to change selectively for identical CS1 trials when the task manipulation only adds a contingency to intertrial crossings.

- Uncertainty also does not increase for CS1 trials, they remain fully predictable and unchanged between conditions.

- Stress is too broad a construct to be meaningful unless clearly operationalized; moreover, any stress differences that arise from task structure would covary with caution rather than replace the interpretation.

(4) Clarifying “types” of responses

The reviewer’s question about “response types” appears to conflate behavioral latencies with the neuronal response “types” defined in the manuscript. The term “type” in this paper refers to neuronal activation derived from movement-based clustering, not to distinct behavioral categories of avoidance, which we term modes.

In sum, we interpret increased CS1 latency as “caution” only when performance remains intact and trial structure is identical between conditions; under those criteria, latency reliably reflects additional cognitive evaluation before acting, rather than nonspecific changes in sensory processing, motivation, etc.

Related to the last, I'm struggling to understand the rationale for dividing cells into 'types' based their physiological responses in some experiments.

There is longstanding precedent in systems neuroscience for classifying neurons by their physiological response patterns, because neurons that respond similarly often play similar functional roles. For example, place cells, grid cells, direction cells, in vivo, and regular spiking, burst firing, and tonic firing in vitro are all defined by characteristic activity patterns in response to stimuli rather than anatomy or genetics alone. In the same spirit, our classifications simply reflect clusters of neurons that exhibit similar ΔF/F dynamics around behaviorally relevant events, such as movement sensitivity or avoidance modes. This is a standard analytic approach used in many studies. Thus, our rationale is not arbitrary: the “classes” and “types” arise from data-driven clustering of physiological responses, consistent with widespread practice, and they help reveal functional distinctions within the STN that would otherwise remain obscured.

In several figures the number of subjects used was not described. This is necessary. Also necessary is some assessment of the variability across subjects.

All the results described include the number of animals. To eliminate uncertainty, we now also include this information in figure legends.

The only measure of error shown in many figures relates trial-to-trial or event variability, which is minimal because in many cases it appears that hundreds of trials may have been averaged per animal, but this doesn't provide a strong view of biological variability (i.e., are results consistent across animals?).

The concern appears to stem from a misunderstanding of what the mixed-effects models quantify. The figure panels often show session-averaged traces for clarity, all statistical inferences in the paper are made at the level of animals, not trials. Mixed-effects modeling is explicitly designed for hierarchical datasets such as ours, where many trials are nested within sessions, which are themselves nested within animals.

In our models, animal is the clustering (random) factor, and sessions are nested within animals, so variability across animals is directly estimated and used to compute the population-level effects. This approach is not only appropriate but is the most stringent and widely recommended method for analyzing behavioral and neural data with repeated measures. In other words, the significance tests and confidence intervals already fully incorporate biological variability across animals.

Thus, although hundreds of trials per animal may be illustrated for visualization, the inferences reflect between-animal consistency, not within-animal trial repetition. The fact that the mixed-effects results are robust across animals supports the biological reliability of the findings.

It is not clear if or how spread of expression outside of target STN was evaluated, and if or how or how many mice were excluded due to spread or fiber placements. Inadequate histological validation is presented and neighboring regions that would be difficult to completely avoid, such as paraSTN may be contributing to some of the effects.

The STN is a compact structure with clear anatomical boundaries, and our injections were rigorously validated to ensure targeting specificity. As detailed in the Methods, every mouse underwent histological verification, and injections were quantified using the Brain Atlas Analyzer app (available on OriginLab), which we developed to align serial sections to the Allen Brain Atlas. This approach provides precise, slice-by-slice confirmation of viral spread. We have performed thousands of AAV injections and probe implants in our lab, incorporating over the years highly reliable stereotaxic procedures with multiple depth and angle checks and tools. For this study specifically, fewer than 10% of mice were excluded due to off-target expression or fiber/lesion placement. None of the included cases showed spread into adjacent structures.

Regarding paraSTN: anatomically, paraSTN is a very small extension contiguous with STN. Our study did not attempt to dissociate subregions within STN, and the viral expression patterns we report fall within the accepted boundaries of STN. Importantly, none of our photometry probes or miniscope lenses sampled paraSTN, so contributions from that region are extremely unlikely to account for any of our neural activity results.

Finally, our paper employs five independent loss-of-function approaches—optogenetic inhibition of STN neurons, selective inhibition of STN projections to the midbrain (in two sites: SNr and mRt), and STN lesions (electrolytic and viral). All methods converge on the same conclusion, providing strong evidence that the effects we report arise from manipulation of STN itself rather than from neighboring regions.

Raw example traces are not provided.

We do not think raw traces are useful here. All figures contain average traces to reflect the average activity of the estimated populations, which are already clustered per classes and types.

The timeline of the spontaneous movement and avoidance sessions were not clear, nor the number of events or sessions per animal and how this was set. It is not clear if there was pre-training or habituation, if many or variable sessions were combined per animal, or what the time gaps between sessions was, or if or how any of these parameters might influence interpretation of the results.

As noted, we have enhanced the description of the sessions, including the number of animals and sessions, which are daily and always equal per animals in each group of experiments. The sessions are part of the random effects in the model. In addition, we now include schematics to facilitate understanding of the procedures.

Comments on revised version:The authors removed the optogenetic stimulation experiments, but then also added a lot of new analyses. Overall the scope of their conclusions are essentially unchanged. Part of the eLife model is to leave it to the authors discretion how they choose to present their work. But my overall view of it is unchanged. There are elements that I found clear, well executed, and compelling. But other elements that I found difficult to understand and where I could not follow or concur with their conclusions.

We respectfully disagree with the assertion that the scope of our conclusions remains unchanged. The revised manuscript differs in several fundamental ways:

(1) Removal of all optogenetic excitation experiments

These experiments were a substantial portion of the original manuscript, and their removal eliminated an entire set of claims regarding the causal control of cautious responding by STN excitation. The revised manuscript no longer makes these claims.

(2) Addition of analyses that directly address the reviewers’ central concerns The new analyses using mixed-effects modeling, window-specific covariates, and movement/baseline controls were added precisely because reviewers requested clearer dissociation of sensory, motor, and task-related contributions. These additions changed not only the presentation but the interpretation of the neural signals. We now conclude that STN encodes movement, caution, and aversive signals in separable ways—not that it exclusively or causally regulates caution.

(3) Clear narrowing of conclusions

Our current conclusions are more circumscribed and data-driven than in the original submission. For example, we removed all claims that STN activation “controls caution,” relying instead on loss-of-function data showing that STN is necessary for performing cued avoidance—not for generating cautious latency shifts. This is a substantial conceptual refinement resulting directly from the review process.

(4) Reorganization to improve clarity

Nearly every section has been restructured, including terminology (mode/type/class), figure organization, and explanations of behavioral windows. These revisions were implemented to ensure that readers can follow the logic of the analyses.

We appreciate the reviewer’s recognition that several elements were clear and compelling. For the remaining points they found difficult to understand, we have addressed each one in detail in the response and revised the manuscript accordingly. If there are still aspects that remain unclear, we would welcome explicit identification of those points so that we can clarify them further.

**Recommendations for the authors:**

**Reviewer #2 (Recommendations for the authors):**
(1) Show individual data points on bar plots- partially addressed. Individual data points are still not shown.

Wherever feasible, we display individual data points (e.g., Figures 1 and 2) to convey variability directly. However, in cases where figures depict hundreds of paired (repeatedmeasures) data points, showing all points without connecting them would not be appropriate, while linking them would make the figures visually cluttered and uninterpretable. All plots and traces include measures of variability (SEM), and the raw data will be shared on Dryad. When error bars are not visible, they are smaller than the trace thickness or bar line—for example, in Figure 5B, the black circles and orange triangles include error bars, but they are smaller than the symbol size.

Also, to minimize visual clutter, only a subset of relevant comparisons is highlighted with asterisks, whereas all relevant statistical results, comparisons, and mouse/session numbers are fully reported in the Results section, with statistical analyses accounting for the clustering of data within subjects and sessions.

(2) The active avoidance experiments are confusing when they are introduced in the results section. More explanation of what paradigms were used and what each CS means at the time these are introduced would add clarity. For example AA1, AA2 etc are explained only with references to other papers, but a brief description of each protocol and a schematic figure would really help.- partially addressed. A schematic figure showing the timeline would still be helpful.

As suggested, we have added an additional panel to Fig. 5A with a schematic describing

AA1-3 tasks. In addition, the avoidance protocols are described briefly but clearly in the Results section (second paragraph of “STN neurons activate during goal-directed avoidance contingencies”) and in greater detail in the Methods section. As stated, these tasks were conducted sequentially, and mice underwent the same number of sessions per procedure, which are indicated. All relevant procedural information has been included in these sections. Mice underwent daily sessions and learnt these tasks within 1-2 sessions, progressing sequentially across tasks with an equal number of sessions per task (7 per task), and the resulting data were combined and clustered by mouse/session in the statistical models.

(3) How do the Class 1, 2, 3 avoids relate to Class 1 , 2, 3 neural types established in Figure 3? It seems like they are not related, and if that is the case they should be named something different from each other to avoid confusion.-not sufficiently addressed. The new naming system of neural 'classes' and 'types' helps with understanding that these are completely different ways of separating subpopulations within the STN. However, it is still unclear why the authors re-type the neurons based on their relation to avoids, when they classify the neurons based on their relationship to speed earlier. And it is unclear whether these neural classes and neural types have anything to do with each other. Are the neural Types related to the neural classes in any way? and what is the overlap between neural types vs classes? Which separation method is more useful for functionally defining STN populations?

The remaining confusion stems from treating several independent analyses as if they were different versions of the same classification. In reality, each analysis asks a distinct question, and the resulting groupings are not expected to overlap or correspond. We clarify this explicitly below.

- Movement onset neuron classes (Class A, B, C; Fig. 3):

These classes categorize neurons based on how their ΔF/F changes around spontaneous movement onset. This analysis identifies which neurons encode the initiation and direction of movement. For instance, Class B neurons (15.9%) were inhibited as movement slowed before onset but did not show sharp activation at onset, whereas Class C neurons (27.6%) displayed a pronounced activation time-locked to movement initiation. Directional analyses revealed that Class C neurons discharged strongly during contraversive turns, while Class B neurons showed a weaker ipsiversive bias. Because neurons were defined per session and many of these recordings did not include avoidance-task sessions, these movement-onset classes were not used in the avoidance analyses.

- Movement-sensitivity neuron classes (Class 1, 2, 3, 4; Fig. 7):

These classes categorize neurons based on the cross-correlation between ΔF/F and head speed, capturing how each neuron’s activity scales with movement features across the entire recording session. This analysis identifies neurons that are strongly speed-modulated, weakly speed-modulated, or largely insensitive to movement. These movement-sensitivity classes were then carried forward into the avoidance analyses to ask how neurons with different kinematic relationships participate during task performance; for example, whether neurons that are insensitive to movement nonetheless show strong activation during avoidance actions.

- Avoidance modes (Mode 1, 2, 3; Fig. 8)

Here we classify actions, not neurons. K-means clustering is applied to the movementspeed time series during CS1 active avoidance trials only, which allows us to identify distinct action modes or variants—fast-onset versus delayed avoidance responses. This action-based classification ensures that we compare neural activity across identical movements, eliminating a major confound in studies that do not explicitly separate action variants. First, we examine how population activity differs across these avoidance modes, reflecting neural encoding of the distinct actions themselves. Second, within each mode, we then classify neurons into “types,” which simply describes how different neurons activate during that specific avoidance action (as noted next).

- Neuron activation types within each mode (Type a, b, c; Fig.9)

This analysis extends the mode-based approach by classifying neuronal activation patterns only within each specific avoidance mode. For each mode, we apply k-means clustering to the ΔF/F time series to identify three activation types—e.g., neurons showing little or no response, neurons showing moderate activation, and neurons showing strong or sharply timed activation. Because all trials within a mode have identical movement profiles, these activation types capture the variability of neural responses to the same avoidance behavior. Importantly, these activation “types” (a, b, c) are not global neuron categories. They do not correspond to, nor are they intended to map onto, the movement-based neuron classes defined earlier. Instead, they describe how neurons differ in their activation during a particular behavioral mode—that is, within a specific set of behaviorally matched trials. Because modes are defined at the trial level, the neurons contributing to each mode can differ: some neurons have trials belonging to one mode, others to two or all three. Thus, Type a/b/c groupings are not fixed properties of neurons. To prevent confusion, we refer to them explicitly as neuronal activation types, emphasizing that they characterize mode-specific response patterns rather than global cell identities.

In conclusion, the categorizations serve entirely different analytical purposes and should not be interpreted as competing classifications. The mode-specific “types” do not reclassify or replace the movement-sensitivity classes; they capture how neurons differ within a single, well-defined avoidance action, while the movement classes reflect how neurons relate to movements in general. Each classification relates to different set of questions and overlap between them is not expected.

To make this as clear as possible we added the following paragraph to the Results:

“To avoid confusion between analyses, it is important to note that the movement-sensitivity classes defined here (Class 1–4; Fig. 7) are conceptually distinct from both the movementonset classes (Class A–C; Fig. 3) and the neuronal activation “types” introduced later in the avoidance-mode analysis. The Class 1–4 grouping reflects how neurons relate to movement across the entire session, based on their cross-correlation with speed. The onset classes A–C capture neural activity specifically around spontaneous movement initiation during general exploration. In contrast, the later activation “types” are derived within each avoidance mode and describe how neurons differ in their activation patterns during identical CS1 avoidance responses. These classifications answer different questions about STN function and are not intended to correspond to one another.”

(4) Similarly having 3 different cell types (a,b,c) in the active avoidance seems unrelated to the original classification of cell types (1,2,3), and these are different for each class of avoid. This is very confusing and it is unclear how any of these types relate to each other. Presumable the same mouse has all three classes of avoids, so there are recording from each cell during each type of avoid. So the authors could compare one cell during each avoid and determine whether it relates to movement or sound or something else. It is interesting that types a,b,c have the exact same proportions in each class of avoid, and really makes it important to investigate if these are the exact same cells or not. Also, these mice could be recorded during open field so the original neural classification (class 1, 2,3) could be applied to these same cells and then the authors can see whether each cell type defined in the open field has different response to the different avoid types. As it stands, the paper simply finds that during movement and during avoidance behaviors different cells in the STN do different things. - Similarly, the authors somewhat addressed the neural types issue, but figure 9 still has 9 different neural types and it is unclear whether the same cells that are type 'a' in mode 1 avoids are also type 'a' in mode 2 avoids, or do some switch to type b? Is there consistency between cell types across avoid modes? The authors show that type 'c' neurons are differentially elevated in mode 3 vs 2, but also describes neurons as type '2c' and statistically compare them to type '1c' neurons. Are these the same neurons? or are type 2c neurons different cells vs type 1c neurons? This is still unclear and requires clarification to be interpretable.

We believe the remaining confusion arises from treating the different classification schemes as if they were alternative labels applied to the same neurons, when in fact they serve entirely separate analytical purposes and may not include the same neurons (see previous point). Because these classifications answer different questions, they are not expected to overlap, nor is overlap required for the interpretations we draw. It is therefore not appropriate to compare a neuron’s “type” in one avoidance mode to its movement class, or to ask whether types a/b/c across different modes are “the same cells,” since modes are defined by trial-level movement clustering rather than by neuron identity. Importantly, Types a/b/c are not intended as a new global classification of neurons; they simply summarize the variability of neuronal responses within each behaviorally matched mode. We agree that future studies could expand our findings, but that is beyond the already wide scope of the present paper. Our current analyses demonstrate a key conceptual point: when movement is held constant (via modes), STN neurons still show heterogeneous, outcome- and caution-related patterns, indicating encoding that cannot be reduced to movement alone.

Relatedly, was the association with speed used to define each neural "class" done in the active avoidance context or in a separate (e.g. open field) experiment? This is not clear in the text.

The cross-correlation classes were derived from the entire recording session, which included open-field and avoidance tasks recordings. The tasks include long intertrial periods with spontaneous movements. We found no difference in classes when we include only a portion of the session, such as the open field or if we exclude the avoidance interval where actions occur.

Finally, in figure 7, why is there a separate avoid trace for each neural class? With the GRIN lens, the authors are presumably getting a sample of all cell types during each avoid, so why do the avoids differ depending on the cell type recorded?

The entire STN population is not recorded within a single session; each session contributes only a subset of neurons to the dataset. Consequently, each neural class is composed of neurons drawn from partially non-overlapping sets of sessions, each with its own movement traces. For this reason, we plot avoidance traces separately for each neural class to maintain strict within-session correspondence between neural activity and the behavior collected in the same sessions. This prevents mixing behavioral data across sessions that did not contribute neurons to that class and ensures that all neural– behavioral comparisons remain appropriately matched. We have clarified this rationale in the revised manuscript. We note that averaging movement across classes—as is often done—would obscure these distinctions and would not preserve the necessary correspondence between neural activity and behavior. This is also clarified in Results.

(5) The use of the same colors to mean two different things in figure 9 is confusing. AA1 vs AA2 shouldn't be the same colors as light-naïve vs light signaling CS.-addressed, but the authors still sometimes use the same colors to mean different things in adjacent figures (e.g. the red, blue, black colors in figure 1 and figure 2 mean totally different things) and use different colors within the same figure to represent the same thing (Figure 9AB vs Figure 9CD). This is suboptimal.

Following the reviewer’s suggestion, in Figure 2, we changed the colors, so readers do not assume they are related to Fig. 1.

In Figure 9, we changed the colors in C,D to match the colors in A,B.

(6) The exact timeline of the optogenetics experiments should be presented as a schematic for understandability. It is not clear which conditions each mouse experienced in which order. This is critical to the interpretation of figure 9 and the reduction of passive avoids during STN stimulation. Did these mice have the CS1+STN stimulation pairing or the STN+US pairing prior to this experiment? If they did, the stimulation of the STN could be strongly associated with either punishment or with the CS1 that predicts punishment. If that is the case, stimulating the STN during CS2 could be like presenting CS1+CS2 at the same time and could be confusing. The authors should make it clear whether the mice were naïve during this passive avoid experiment or whether they had experienced STN stimulation paired with anything prior to this experiment.-addressed(7) Similarly, the duration of the STN stimulation should be made clear on the plots that show behavior over time (e.g. Figure 9E).-addressed(8) There is just so much data and so many conditions for each experiment here. The paper is dense and difficult to read. It would really benefit readability if the authors put only the key experiments and key figure panels in the main text and moved much of the repetative figure panels to supplemental figures. The addition of schematic drawings for behavioral experiment timing and for the different AA1, AA2, AA3 conditions would also really improve clarity.-partially addressed. The paper is still dense and difficult to read. No experimental schematics were added.

As suggested, we now added the schematic to Fig. 5A.

New Comments:(9) Description of the animals used and institutional approval are missing from the methods.

The information on animal strains and institutional approval is already included in the manuscript. The first paragraph of the Methods section states:

“… All procedures were reviewed and approved by the institutional animal care and use committee and conducted in adult (>8 weeks) male and female mice. …”

Additionally, the next subsection, “Strains and Adeno-Associated Viruses (AAVs),” fully specifies all mouse lines used. We therefore believe that the required descriptions of animals and institutional approval are already present and meet standard reporting.